# Large Scale Distributed Sparse Precision Estimation

**Huahua Wang, Arindam Banerjee**
Dept. of Computer Science & Engg, University of Minnesota, Twin Cities
{huwang,banerjee}@cs.umn.edu

**Cho-Jui Hsieh, Pradeep Ravikumar, Inderjit S. Dhillon**
Dept. of Computer Science, University of Texas, Austin
{cjhsieh,pradeepr,inderjit}@cs.utexas.edu

## Abstract

We consider the problem of sparse precision matrix estimation in high dimensions using the CLIME estimator, which has several desirable theoretical properties. We present an inexact alternating direction method of multiplier (ADMM) algorithm for CLIME, and establish rates of convergence for both the objective and optimality conditions. Further, we develop a large scale distributed framework for the computations, which scales to millions of dimensions and trillions of parameters, using hundreds of cores. The proposed framework solves CLIME in column-blocks and only involves elementwise operations and parallel matrix multiplications. We evaluate our algorithm on both shared-memory and distributed-memory architectures, which can use block cyclic distribution of data and parameters to achieve load balance and improve the efficiency in the use of memory hierarchies. Experimental results show that our algorithm is substantially more scalable than state-of-the-art methods and scales almost linearly with the number of cores.

## 1 Introduction

Consider a $p$-dimensional probability distribution with true covariance matrix $\Sigma_0 \in \mathcal{S}_{++}^p$ and true precision (or inverse covariance) matrix $\Omega_0 = \Sigma_0^{-1} \in \mathcal{S}_{++}^p$. Let $[R_1 \ \cdots \ R_n] \in \Re^{p \times n}$ be $n$ independent and identically distributed random samples drawn from this $p$-dimensional distribution. The centered normalized sample matrix $\mathbf{A} = [\mathbf{a}_1 \ \cdots \mathbf{a}_n] \in \Re^{p \times n}$ can be obtained as $\mathbf{a}_i = \frac{1}{\sqrt{n}}(R_i - \bar{R})$, where $\bar{R} = \frac{1}{n}\sum_i R_i$, so that the sample covariance matrix can be computed as $\mathbf{C} = \mathbf{A}\mathbf{A}^T$. In recent years, considerable effort has been invested in obtaining an accurate estimate of the precision matrix $\hat{\Omega}$ based on the sample covariance matrix $\mathbf{C}$ in the 'low sample, high dimensions' setting, i.e., $n \ll p$, especially when the true precision $\Omega_0$ is assumed to be sparse [28]. Suitable estimators and corresponding statistical convergence rates have been established for a variety of settings, including distributions with sub-Gaussian tails, polynomial tails [25, 3, 19]. Recent advances have also established parameter-free methods which achieve minimax rates of convergence [4, 19].

Spurred by these advances in the statistical theory of precision matrix estimation, there has been considerable recent work on developing computationally efficient *optimization methods* for solving the corresponding statistical estimation problems: see [1, 8, 14, 21, 13], and references therein. While these methods are able to efficiently solve problems up to a few thousand variables, ultra-large-scale problems with millions of variables remain a challenge. Note further that in precision matrix estimation, the number of parameters scales quadratically with the number of variables; so that with a million dimensions $p = 10^6$, the total number of parameters to be estimated is a trillion, $p^2 = 10^{12}$. The focus of this paper is on designing an efficient distributed algorithm for precision matrix estimation under such ultra-large-scale dimensional settings.

We focus on the CLIME statistical estimator [3], which solves the following linear program (LP):

$$\min \|\hat{\Omega}\|_1 \quad \text{s.t.} \quad \|\mathbf{C}\hat{\Omega} - \mathbf{I}\|_\infty \leq \lambda \,, \tag{1}$$

where $\lambda > 0$ is a tuning parameter. The CLIME estimator not only has strong statistical guarantees [3], but also comes with inherent computational advantages. First, the LP in (1) does not explicitly enforce positive definiteness of $\hat{\Omega}$, which can be a challenge to handle efficiently in high-dimensions. Secondly, it can be seen that (1) can be decomposed into $p$ independent LPs, one for each column of $\hat{\Omega}$. This separable structure has motivated solvers for (1) which solve the LP column-by-column using interior point methods [3, 28] or the alternating direction method of multipliers (ADMM) [18]. However, these solvers do not scale well to ultra-high-dimensional problems: they are not designed to run on hundreds to thousands of cores, and in particular require the entire sample covariance matrix $\mathbf{C}$ to be loaded into the memory of a single machine, which is impractical even for moderate sized problems.

In this paper, we present an efficient CLIME-ADMM variant along with a scalable distributed framework for the computations [2, 26]. The proposed CLIME-ADMM algorithm can scale up to millions of dimensions, and can use up to thousands of cores in a shared-memory or distributed-memory architecture. The scalability of our method relies on the following key innovations. First, we propose an inexact ADMM [27, 12] algorithm targeted to CLIME, where each step is either elementwise parallel or involves suitable matrix multiplications. We show that the rates of convergence of the objective to the optimum as well as residuals of constraint violation are both $O(1/T)$. Second, we solve (1) in column-blocks of the precision matrix at a time, rather than one column at a time. Since (1) already decomposes columnwise, solving multiple columns together in blocks might not seem worthwhile. However, as we show our CLIME-ADMM working with column-blocks uses matrix-matrix multiplications which, building on existing literature [15, 5, 11] and the underlying low rank and sparse structure inherent in the precision matrix estimation problem, can be made substantially more efficient than repeated matrix-vector multiplications. Moreover, matrix multiplication can be further simplified as block-by-block operations, which allows choosing optimal block sizes to minimize cache misses, leading to high scalability and performance [16, 5, 15]. Lastly, since the core computations can be parallelized, CLIME-ADMM scales almost linearly with the number of cores. We experiment with shared-memory and distributed-memory architectures to illustrate this point. Empirically, CLIME-ADMM is shown to be much faster than existing methods for precision estimation, and scales well to high-dimensional problems, e.g., we estimate a precision matrix of one million dimension and one trillion parameters in 11 hours by running the algorithm on 400 cores.

Our framework can be positioned as a part of the recent surge of effort in scaling up machine learning algorithms [29, 22, 6, 7, 20, 2, 23, 9] to "Big Data". Scaling up machine learning algorithms through parallelization and distribution has been heavily explored on various architectures, including shared-memory architectures [22], distributed memory architectures [23, 6, 9] and GPUs [24]. Since MapReduce [7] is not efficient for optimization algorithms, [6] proposed a parameter server that can be used to parallelize gradient descent algorithms for unconstrained optimization problems. However, this framework is ill-suited for the constrained optimization problems we consider here, because gradient descent methods require the projection at each iteration which involves all variables and thus ruins the parallelism. In other recent related work based on ADMM, [23] introduce graph projection block splitting (GPBS) to split data into blocks so that examples and features can be distributed among multiple cores. Our framework uses a more general blocking scheme (block cyclic distribution), which provides more options in choosing the optimal block size to improve the efficiency in the use of memory hierarchies and minimize cache misses [16, 15, 5]. ADMM has also been used to solve constrained optimization in a distributed framework [9] for graphical model inference, but they consider local constraints, in contrast to the global constraints in our framework.

**Notation:** A matrix is denoted by a bold face upper case letter, e.g., $\mathbf{A}$. An element of a matrix is denoted by a upper case letter with row index $i$ and column index $j$, e.g., $A_{ij}$ is the $ij$-th element of $\mathbf{A}$. A block of matrix is denoted by a bold face lower case letter indexed by $ij$, e.g., $\mathbf{A}_{ij}$. $\vec{\mathbf{A}}_{ij}$ represents a collection of blocks of matrix $\mathbf{A}$ on the $ij$-th core (see block cyclic distribution in Section 4). $\mathbf{A}'$ refers the transpose of $\mathbf{A}$. Matrix norms used are all elementwise norms, e.g., $\|\mathbf{A}\|_1 = \sum_{i=1}^{p} \sum_{j=1}^{n} |A_{ij}|$, $\|\mathbf{A}\|_2^2 = \sum_{i=1}^{p} \sum_{j=1}^{n} A_{ij}^2$, $\|\mathbf{A}\|_\infty = \max_{1 \le i \le p, 1 \le j \le n} |A_{ij}|$. The matrix inner product is defined in elementwise, e.g., $\langle \mathbf{A}, \mathbf{B} \rangle = \sum_{i=1}^{p} \sum_{j=1}^{n} A_{ij} B_{ij}$. $\mathbf{X} \in \Re^{p \times k}$ denotes $k(1 \le k \le p)$ columns of the precision matrix $\hat{\Omega}$, and $\mathbf{E} \in \Re^{p \times k}$ denotes the same $k$ columns of the identity matrix $\mathbf{I} \in \Re^{p \times p}$. Let $\lambda_{\max}(\mathbf{C})$ be the largest eigenvalue of covariance matrix $\mathbf{C}$.

---

**Algorithm 1** Column Block ADMM for CLIME

---

1: **Input:** $\mathbf{C}, \lambda, \rho, \eta$
2: **Output:** $\mathbf{X}$
3: **Initialization:** $\mathbf{X}^0, \mathbf{Z}^0, \mathbf{Y}^0, \mathbf{V}^0, \hat{\mathbf{V}}^0 = 0$
4: **for** $t = 0$ to $T - 1$ **do**
5:   **X-update:** $\mathbf{X}^{t+1} = soft(\mathbf{X}^t - \mathbf{V}^t, \frac{1}{\eta})$, where $\quad soft(\mathbf{X}, \gamma) = \begin{cases} X_{ij} - \gamma\,, & \text{if } X_{ij} > \gamma\,, \\ X_{ij} + \gamma\,, & \text{if } X_{ij} < -\gamma\,, \\ 0\,, & \text{otherwise} \end{cases}$
6:   **Mat-Mul:** $\begin{cases} \text{sparse}: & \mathbf{U}^{t+1} = \mathbf{C}\mathbf{X}^{t+1} \\ \text{low rank}: & \mathbf{U}^{t+1} = \mathbf{A}(\mathbf{A}'\mathbf{X}^{t+1}) \end{cases}$
7:   **Z-update:** $\mathbf{Z}^{t+1} = box(\mathbf{U}^{t+1} + \mathbf{Y}^t, \lambda)$, where $box(\mathbf{X}, \mathbf{E}, \lambda) = \begin{cases} E_{ij} + \lambda, & \text{if } X_{ij} - E_{ij} > \lambda, \\ X_{ij}, & \text{if } |X_{ij} - E_{ij}| \leq \lambda, \\ E_{ij} - \lambda, & \text{if } X_{ij} - E_{ij} < -\lambda, \end{cases}$
8:   **Y-update:** $\mathbf{Y}^{t+1} = \mathbf{Y}^t + \mathbf{U}^{t+1} - \mathbf{Z}^{t+1}$
9:   **Mat-Mul:** $\begin{cases} \text{sparse}: & \hat{\mathbf{V}}^{t+1} = \mathbf{C}\mathbf{Y}^{t+1} \\ \text{low rank}: & \hat{\mathbf{V}}^{t+1} = \mathbf{A}(\mathbf{A}'\mathbf{Y}^{t+1}) \end{cases}$
10:   **V-update:** $\mathbf{V}^{t+1} = \frac{\rho}{\eta}(2\hat{\mathbf{V}}^{t+1} - \hat{\mathbf{V}}^t)$
11: **end for**

---

## 2 Column Block ADMM for CLIME

In this section, we propose an algorithm to estimate the precision matrix in terms of column blocks instead of column-by-column. Assuming a column block contains $k(1 \leq k \leq p)$ columns, the sparse precision matrix estimation amounts to solving $\lceil p/k \rceil$ independent linear programs. Denoting $\mathbf{X} \in \Re^{p \times k}$ be $k$ columns of $\hat{\Omega}$, (1) can be written as

$$\min \|\mathbf{X}\|_1 \quad \text{s.t.} \quad \|\mathbf{C}\mathbf{X} - \mathbf{E}\|_\infty \leq \lambda\,, \tag{2}$$

which can be rewritten in the following equality-constrained form:

$$\min \|\mathbf{X}\|_1 \quad \text{s.t.} \quad \|\mathbf{Z} - \mathbf{E}\|_\infty \leq \lambda, \mathbf{C}\mathbf{X} = \mathbf{Z}\,. \tag{3}$$

Through the splitting variable $\mathbf{Z} \in \Re^{p \times k}$, the infinity norm constraint becomes a box constraint and is separated from the $\ell_1$ norm objective. We use ADMM to solve (3). The augmented Lagrangian of (3) is

$$L_\rho = \|\mathbf{X}\|_1 + \rho\langle \mathbf{Y}, \mathbf{C}\mathbf{X} - \mathbf{Z}\rangle + \frac{\rho}{2}\|\mathbf{C}\mathbf{X} - \mathbf{Z}\|_2^2\,, \tag{4}$$

where $\mathbf{Y} \in \Re^{p \times k}$ is a scaled dual variable and $\rho > 0$. ADMM yields the following iterates [2]:

$$\mathbf{X}^{t+1} = \text{argmin}_{\mathbf{X}} \|\mathbf{X}\|_1 + \frac{\rho}{2}\|\mathbf{C}\mathbf{X} - \mathbf{Z}^t + \mathbf{Y}^t\|_2^2\,, \tag{5}$$

$$\mathbf{Z}^{t+1} = \underset{\|\mathbf{Z} - \mathbf{E}\|_\infty \leq \lambda}{\text{argmin}} \frac{\rho}{2}\|\mathbf{C}\mathbf{X}^{t+1} - \mathbf{Z} + \mathbf{Y}^t\|_2^2\,, \tag{6}$$

$$\mathbf{Y}^{t+1} = \mathbf{Y}^t + \mathbf{C}\mathbf{X}^{t+1} - \mathbf{Z}^{t+1}\,. \tag{7}$$

As a Lasso problem, (5) can be solved using exisiting Lasso algorithms, but that will lead to a double-loop algorithm. (5) does not have a closed-form solution since $C$ in the quadratic penalty term makes $\mathbf{X}$ coupled. We decouple $\mathbf{X}$ by linearizing the quadratic penalty term and adding a proximal term as follows:

$$\mathbf{X}^{t+1} = \text{argmin}_{\mathbf{X}} \|\mathbf{X}\|_1 + \eta\langle \mathbf{V}^t, \mathbf{X}\rangle + \frac{\eta}{2}\|\mathbf{X} - \mathbf{X}^t\|_2^2\,, \tag{8}$$

where $\mathbf{V}^t = \frac{\rho}{\eta}\mathbf{C}(\mathbf{Y}^t + \mathbf{C}\mathbf{X}^t - \mathbf{Z}^t)$ and $\eta > 0$. (8) is usually called an inexact ADMM update. Using (7), $\mathbf{V}^t = \frac{\rho}{\eta}\mathbf{C}(2\mathbf{Y}^t - \mathbf{Y}^{t-1})$. Let $\hat{\mathbf{V}}^t = \mathbf{C}\mathbf{Y}^t$, we have $\mathbf{V}^t = \frac{\rho}{\eta}(2\hat{\mathbf{V}}^t - \hat{\mathbf{V}}^{t-1})$. (8) has the following closed-form solution:

$$\mathbf{X}^{t+1} = soft(\mathbf{X}^t - \mathbf{V}^t, \frac{1}{\eta})\,, \tag{9}$$

where *soft* denotes the soft-thresholding and is defined in Step 5 of Algorithm 1.

Let $\mathbf{U}^{t+1} = \mathbf{C}\mathbf{X}^{t+1}$. (6) is a box constrained quadratic programming which has the following closed-form solution:

$$\mathbf{Z}^{t+1} = box(\mathbf{U}^{t+1} + \mathbf{Y}^t, \mathbf{E}, \lambda)\,, \tag{10}$$

where *box* denotes the projection onto the infinity norm constraint $\|\mathbf{Z} - \mathbf{E}\|_\infty \leq \lambda$ and is defined in Step 7 of Algorithm 1. In particular, if $\|\mathbf{U}^{t+1} + \mathbf{Y}^t - \mathbf{E}\|_\infty \leq \lambda$, $\mathbf{Z}^{t+1} = \mathbf{U}^{t+1} + \mathbf{Y}^t$ and thus $\mathbf{Y}^{t+1} = \mathbf{Y}^t + \mathbf{U}^{t+1} - \mathbf{Z}^{t+1} = \mathbf{0}$.

The ADMM algorithm for CLIME is summarized in Algorithm 1. In Algorithm 1, while step 5, 7, 8 and 10 amount to elementwise operations which cost $O(pk)$ operations, steps 6 and 9 involve matrix multiplication which is the most computationally intensive part and costs $O(p^2 k)$ operations. The memory requirement includes $O(pn)$ for $\mathbf{A}$ and $O(pk)$ for the other six variables.

As the following results show, Algorithm 1 has a $O(1/T)$ convergence rate for both the objective function and the residuals of optimality conditions. The proof technique is similar to [26]. [12] shows a similar result as Theorem 2 but uses a different proof technique. For proofs, please see Appendix A in the supplement.

**Theorem 1** *Let $\{\mathbf{X}^t, \mathbf{Z}^t, \mathbf{Y}^t\}$ be generated by Algorithm 1 and $\bar{\mathbf{X}}^T = \frac{1}{T}\sum_{t=1}^T \mathbf{X}^t$. Assume $\mathbf{X}^0 = \mathbf{Z}^0 = \mathbf{Y}^0 = \mathbf{0}$ and $\eta \geq \rho\lambda_{\max}^2(\mathbf{C})$. For any $\mathbf{CX} = \mathbf{Z}$, we have*

$$\|\bar{\mathbf{X}}^T\|_1 - \|\mathbf{X}\|_1 \leq \frac{\eta\|\mathbf{X}\|_2^2}{2T} \ . \tag{11}$$

**Theorem 2** *Let $\{\mathbf{X}^t, \mathbf{Z}^t, \mathbf{Y}^t\}$ be generated by Algorithm 1 and $\{\mathbf{X}^*, \mathbf{Z}^*, \mathbf{Y}^*\}$ be a KKT point for the Lagrangian of (3). Assume $\mathbf{X}^0 = \mathbf{Z}^0 = \mathbf{Y}^0 = \mathbf{0}$ and $\eta \geq \rho\lambda_{\max}^2(\mathbf{C})$. We have*

$$\|\mathbf{CX}^T - \mathbf{Z}^T\|_2^2 + \|\mathbf{Z}^T - \mathbf{Z}^{T-1}\|_2^2 + \|\mathbf{X}^T - \mathbf{X}^{T-1}\|_{\frac{\eta}{\rho}\mathbf{I} - \mathbf{C}^2}^2 \leq \frac{\|\mathbf{Y}^*\|_2^2 + \frac{\eta}{\rho}\|\mathbf{X}^*\|_2^2}{T} \ . \tag{12}$$

## 3 Leveraging Sparse, Low-Rank Structure

In this section, we consider a few possible directions that can further leverage the underlying structure of the problem; specifically sparse and low-rank structure.

### 3.1 Sparse Structure

As we detail here, there could be sparsity in the intermediate iterates, or the sample covariance matrix itself (or a perturbed version thereof); which can be exploited to make our CLIME-ADMM variant more efficient.

**Iterate Sparsity:** As the iterations progress, the soft-thresholding operation will yield a sparse $\mathbf{X}^{t+1}$, which can help speed up step 6: $\mathbf{U}^{t+1} = \mathbf{C}X^{t+1}$, via sparse matrix multiplication. Further, the box-thresholding operation will yield a sparse $\mathbf{Y}^{t+1}$. In the ideal case, if $\|\mathbf{U}^{t+1} + \mathbf{Y}^t - \mathbf{E}\|_\infty \leq \lambda$ in step 7, then $\mathbf{Z}^{t+1} = \mathbf{U}^{t+1} + \mathbf{Y}^t$. Thus, $\hat{\mathbf{Y}}^{t+1} = \mathbf{Y}^t + \mathbf{U}^{t+1} - \mathbf{Z}^{t+1} = \mathbf{0}$. More generally, $\mathbf{Y}^{t+1}$ will become sparse as the iterations proceed, which can help speed up step 9: $\hat{\mathbf{V}}^{t+1} = \mathbf{C}\mathbf{Y}^{t+1}$.

**Sample Covariance Sparsity:** We show that one can "perturb" the sample covariance to obtain a sparse and coarsened matrix, solve CLIME with this pertubed matrix, and yet have strong statistical guarantees. The statistical guarantees for CLIME [3], including convergence in spectral, matrix $L_1$, and Frobenius norms, only require from the sample covariance matrix $\mathbf{C}$ a deviation bound of the form $\|\mathbf{C} - \Sigma_0\|_\infty \leq c\sqrt{\log p/n}$, for some constant $c$. Accordingly, if we perturb the matrix $\mathbf{C}$ with a perturbation matrix $\Delta$ so that the perturbed matrix $(\mathbf{C} + \Delta)$ continues to satisfy the deviation bound, the statistical guarantees for CLIME would hold even if we used the perturbed matrix $(\mathbf{C} + \Delta)$. The following theorem (for details, please see Appendix B in the supplement) illustrates some perturbations $\Delta$ that satisfy this property:

**Theorem 3** *Let the original random variables $R_i$ be sub-Gaussian, with sample covariance $\mathbf{C}$. Let $\Delta$ be a random perturbation matrix, where $\Delta_{ij}$ are independent sub-exponential random variables. Then, for positive constants $c_1, c_2, c_3$, $P(\|\mathbf{C} + \Delta - \Sigma_0\|_\infty \geq c_1\sqrt{\frac{\log p}{n}}) \leq c_2 p^{-c_3}$.*

As a special case, one can thus perturb elements of $C_{ij}$ with suitable constants $\Delta_{ij}$ with $|\Delta_{ij}| \leq c\sqrt{\log p/n}$, so that the perturbed matrix is sparse, i.e., if $|C_{ij}| \leq c\sqrt{\log p/n}$, then it can be safely

truncated to 0. Thus, in practice, even if sample covariance matrix is only close to a sparse matrix [21, 13], or if it is close to being block diagonal [21, 13], the complexity of matrix multiplication in steps 6 and 9 can be significantly reduced via the above perturbations.

## 3.2 Low Rank Structure

Although one can use sparse structures of matrices participating in the matrix multiplication to accelerate the algorithm, the implementation requires substantial work since dynamic sparsity of $\mathbf{X}$ and $\mathbf{Y}$ is unknown upfront and static sparsity of the sample covariance matrix may not exist. Since the method will operate in a low-sample setting, we can alternatively use the *low rank* of the sample covariance matrix to reduce the complexity of matrix multiplication. Since $\mathbf{C} = \mathbf{A}\mathbf{A}^T$ and $p \gg n$, $\mathbf{C}\mathbf{X} = \mathbf{A}(\mathbf{A}^T\mathbf{X})$, and thus the computational complexity of matrix multiplication reduces from $O(p^2 k)$ to $O(npk)$, which can achieve significant speedup for small $n$. We use such low-rank multiplications for the experiments in Section 5.

# 4 Scalable Parallel Computation Framework

In this section, we elaborate on scalable frameworks for CLIME-ADMM in both shared-memory and distributed-memory achitectures.

In a shared-memory architecture (e.g., a single machine), data $\mathbf{A}$ is loaded to the memory and shared by $q$ cores, as shown in Figure 1(a). Assume the $p \times p$ precision matrix $\hat{\Omega}$ is evenly divided into $l = p/k \ (\geq q)$ column blocks, e.g., $\mathbf{X}^1, \cdots, \mathbf{X}^q, \cdots, \mathbf{X}^l$, and thus each column block contains $k$ columns. The column blocks are assigned to $q$ cores cyclically, which means the $j$-th column block is assigned to the $mod(j, q)$-th core. The $q$ cores can solve $q$ column blocks in parallel without communication and synchronization, which can be simply implemented via multithreading. Meanwhile, another $q$ column blocks are waiting in their respective queues. Figure 1(a) gives an example of how to solve 8 column blocks on 4 cores in a shared-memory environment. While the 4 cores are solving the first 4 column blocks, the next 4 column blocks are waiting in queues (red arrows).

Although the shared-memory framework is free from communication and synchronization, the limited resources prevent it from scaling up to datasets with millions of dimensions, which can not be loaded to the memory of a single machine or solved by tens of cores in a reasonble time. As more memory and computing power are needed for high dimensional datasets, we implement a framework for CLIME-ADMM in a distributed-memory architecture, which automatically distributes data among machines, parallelizes computation, and manages communication and synchronization among machines, as shown in Figure 1(b). Assume $q$ processes are formed as a $r \times c$ process grid and the $p \times p$ precision matrix $\hat{\Omega}$ is evenly divided into $l = p/k \ (\geq q)$ column blocks, e.g., $\mathbf{X}^j, 1 \leq j \leq l$. We solve a column block $\mathbf{X}^j$ at a time in the process grid. Assume the data matrix $\mathbf{A}$ has been evenly distributed into the process grid and $\vec{\mathbf{A}}_{ij}$ is the data on the $ij$-th core, i.e., $\mathbf{A}$ is colletion of $\vec{\mathbf{A}}_{ij}$ under a mapping scheme, which we will discuss later. Figure 1(b) illustrates that the $2 \times 2$ process grid is computing the first column block $\mathbf{X}^1$ while the second column block $\mathbf{X}^2$ is waiting in queues (red lines), assuming $\mathbf{X}^1, \mathbf{X}^2$ are distributed into the process grid in the same way as $\mathbf{A}$ and $\vec{\mathbf{X}}_{ij}^1$ is the block of $\mathbf{X}^1$ assigned to the $ij$-th core.

A typical issue in parallel computation is load imbalance, which is mainly caused by the computational disparity among cores and leads to unsatisfactory speedups. Since each step in CLIME-ADMM are basic operations like matrix multiplication, the distribution of sub-matrices over processes has a major impact on the load balance and scalability. The following discussion focuses on the matrix multiplication in the step 6 in Algorithm 1. Other steps can be easily incorporated into the framework. The matrix multiplication $\mathbf{U} = \mathbf{A}(\mathbf{A}'\mathbf{X}^1)$ can be decomposed into two steps, i.e., $\mathbf{W} = \mathbf{A}'\mathbf{X}^1$ and $\mathbf{U} = \mathbf{A}\mathbf{W}$, where $\mathbf{A} \in \Re^{n \times p}$, $\mathbf{X}^1 \in \Re^{p \times k}$, $\mathbf{W} \in \Re^{n \times k}$ and $\mathbf{U} \in \Re^{n \times k}$. Dividing matrices $\mathbf{A}, \mathbf{X}$ evenly into $r \times c$ large consecutive blocks like [23] will lead to load imbalance. First, since the sparse structure of $\mathbf{X}$ changes over time (Section 3.1), large consecutive blocks may assign dense blocks to some processes and sparse blocks to the other processes. Second, there will be no blocks in some processes after the multiplication using large blocks since $\mathbf{W}$ is a small matrix compared to $\mathbf{A}, \mathbf{X}$, e.g., $p$ could be millions and $n, k$ are hundreds. Third, large blocks may not be fit in the cache, leading to cache misses. Therefore, we use block cyclic data distribution which uses a small nonconsecutive blocks and thus can largely achieve load balance and scalability. A matrix is first divided into consecutive blocks of size $p_b \times n_b$. Then blocks are distributed into the process

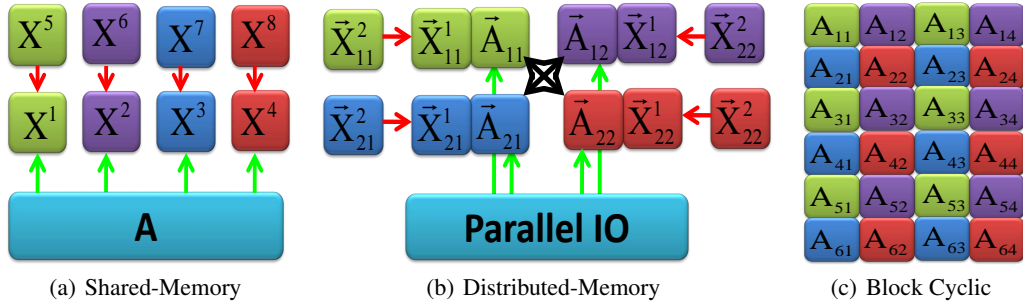

| (a) Shared-Memory | (b) Distributed-Memory | (c) Block Cyclic |

Figure 1: CLIME-ADMM on shared-memory and distribtued-memory architectures.

grid cyclically. Figure 1(c) illustrates how to distribute the matrix to a $2 \times 2$ process grid. $\mathbf{A}$ is divided into $3 \times 2$ consecutive blocks, where each block is of size $p_b \times n_b$. Blocks of the same color will be assigned to the same process. Green blocks will be assigned to the upper left process, i.e., $\vec{\mathbf{A}}_{11} = \{\mathbf{a}_{11}, \mathbf{a}_{13}, \mathbf{a}_{31}, \mathbf{a}_{33}, \mathbf{a}_{51}, \mathbf{a}_{53}\}$ in Figure 1(b). The distribution of $\mathbf{X}^1$ can be done in a similar way except the block size should be $p_b \times k_b$, where $p_b$ is to guarantee that matrix multiplication $\mathbf{A}'\mathbf{X}^1$ works. In particular, we denote $p_b \times n_b \times k_b$ as the block size for matrix multiplication. To distribute the data in a block cyclic manner, we use a parallel I/O scheme, where processes can access the data in parallel and only read/write the assigned blocks.

# 5 Experimental Results

In this section, we present experimental results to compare CLIME-ADMM with existing algorithms and show its scalability. In all experiments, we use the low rank property of the sample covariance matrix and do not assume any other special structures. Our algorithm is implemented in a shared-memory architecture using OpenMP (http://openmp.org/wp/) and a distributed-memory architecture using OpenMPI (http://www.open-mpi.org) and ScaLAPACK [15] (http://www.netlib.org/scalapack/).

## 5.1 Comparision with Existing Algorithms

We compare CLIME-ADMM with three other methods for estimating the inverse covariance matrix, including CLIME, Tiger in package flare[1] and divide and conquer QUIC (DC-QUIC) [13]. The comparisons are run on an Intel Zeon E5540 2.83GHz CPU with 32GB main memory.

We test the efficiency of the above methods on both synthetic and real datasets. For synthetic datasets, we generate the underlying graphs with random nonzero pattern by the same way as in [14]. We control the sparsity of the underlying graph to be $0.05$, and generate random graphs with various dimension. Since each estimator has different parameters to control the sparsity, we set them individually to recover the graph with sparsity $0.05$, and compare the time to get the solution. The column block size $k$ for CLIME-ADMM is 100. Figure 2(a) shows that CLIME-ADMM is the most scalable estimator for large graphs. We compare the precision and recall for different methods on recovering the groud truth graph structure. We run each method using different parameters (which controls the sparsity of the solution), and plot the precision and recall for each solution in Figure 2(b). As Tiger is parameter tuning free and achieves the minimax optimal rate [19], it achieves the best performance in terms of recall. The other three methods have the similar performance. CLIME can also be free of parameter tuning and achieve the optimal minimax rate by solving an additional linear program which is similar to (1) [4]. We refer the readers to [3, 4, 19] for detailed comparisons between the two models CLIME and Tiger, which is not the focus of this paper.

We further test the efficiency of the above algorithms on two real datasets, Leukemia and Climate (see Table 1). Leukemia is gene expression data provided by [10], and the pre-processing was done by [17]. Climate dataset is the temperature data in year 2001 recorded by NCEP/NCAR Reanalysis data[2] and preprocessed by [13]. Since the ground truth for real datasets are unknown, we test the time taken for each method to recover graphs with $0.1$ and $0.01$ sparsity. The results are presented in Table 1. Although Tiger is faster than CLIME-ADMM on small dimensional dataset Leukemia,

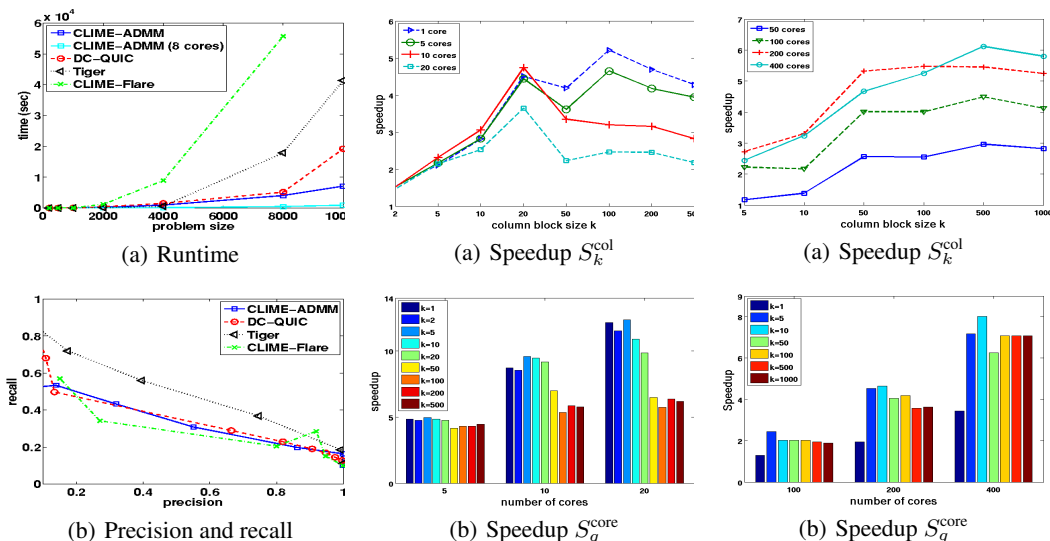

Figure 2: Synthetic datasets     Figure 3: Shared-Memory.     Figure 4: Distributed-Memory.

it does not scale well on the high dimensional dataset as CLIME-ADMM, which is mainly due to the fact that ADMM is not competitive with other methods on small problems but has superior scalability on big datasets [2]. DC-QUIC runs faster than other methods for small sparsity but dramatically slows down when sparsity increases. DC-QUIC essentially works on a block-diagonal matrix by thresholding the off-diagonal elements of the sample covariance matrix. A small sparsity generally leads to small diagonal blocks, which helps DC-QUIC to make a giant leap forward in the computation. A block-diagonal structure in the sample covariance matrix can be easily incorporated into the matrix multiplication in CLIME-ADMM to achieve a sharp computational gain. On a single core, CLIME-ADMM is faster than flare ADMM. We also show the results of CLIME-ADMM on 8 cores, showing CLIME-ADMM achieves a linear speedup (more results will be seen in Section 5.2). Note Tiger can estimate the spase precision matrix column-by-column in parallel, while CLIME-ADMM solves CLIME in column-blocks in parallel.

## 5.2 Scalability of CLIME ADMM

We evaluate the scalability of CLIME-ADMM in a shared memory and a distributed memory architecture in terms of two kinds of speedups. The first speedup is defined as the time on 1 core $T_1^{\text{core}}$ over $q$ cores $T_q^{\text{core}}$, i.e., $S_q^{\text{core}} = T_1^{\text{core}}/T_q^{\text{core}}$. The second speedup is caused by the use of column blocks. Assume the total time for solving CLIME column-by-column ($k = 1$) is $T_1^{\text{col}}$, which is considered as the baseline. The speedup of solving CLIME in column block with size $k$ over a single column is defined as $S_k^{\text{col}} = T_1^{\text{col}}/T_k^{\text{col}}$. The experiments are done on synthetic data which is generated in the same way as in Section 5.1. The number of samples is fixed to be $n = 200$.

**Shared-memory** We estimate a precision matrix with $p = 10^4$ dimensions on a server with 20 cores and 64G memory. We use OpenMP to parallelize column blocks. We run the algorithm on different number of cores $q = 1, 5, 10, 20$, and with different column block size $k$. The speedup $S_k^{\text{col}}$ is plotted in Figure 3(a), which shows the results on three different number of cores. When $k \leq 20$, the speedups keep increasing with increasing number of columns $k$ in each block. For $k \geq 20$, the speedups are maintained on 1 core and 5 cores, but decreases on 10 and 20 cores. The total number of columns in the shared-memory is $k \times q$. For a fixed $k$, more columns are involved in the computation when more cores are used, leading to more memory consumption and competition for the usage of shared cache. The speedup $S_q^{\text{core}}$ is plotted in Figure 3(b), where $T_1^{\text{core}}$ is the time on a single core. The ideal linear speedups are archived on 5 cores for all block sizes $k$. On 10 cores, while small and medium column block sizes can maintain the ideal linear speedups, the large column block sizes fail to scale linearly. The failure to achieve a linear speedup propagate to small and medium column block sizes on 20 cores, although their speedups are larger than large column block size. As more and more column blocks are participating in the computation, the speed-ups decrease possibly because of the competition for resources (e.g., L2 cache) in the shared-memory environment.

Table 1: Comparison of runtime (sec) on real datasets.

| Dataset | sparsity | CLIME-ADMM | | DC-QUIC | Tiger | flare CLIME |
|---|---|---|---|---|---|---|
| | | 1 core | 8 cores | | | |
| Leukemia | 0.1 | 48.64 | 6.27 | 93.88 | 34.56 | 142.5 |
| $(1255 \times 72)$ | 0.01 | 44.98 | 5.83 | 21.59 | 17.10 | 87.60 |
| Climate | 0.1 | 4.76 hours | 0.6 hours | 10.51 hours | > 1 day | > 1 day |
| $(10512 \times 1464)$ | 0.01 | 4.46 hours | 0.56 hours | 2.12 hours | > 1 day | > 1 day |

Table 2: Effect (runtime (sec)) of using different number of cores in a node with $p = 10^6$. Using one core per node is the most efficient as there is no resource sharing with other cores.

| node $\times$ core | k = 1 | k = 5 | k = 10 | k = 50 | k = 100 | k = 500 | k = 1000 |
|---|---|---|---|---|---|---|---|
| 100$\times$1 | 0.56 | 1.26 | 2.59 | 6.98 | 13.97 | 62.35 | 136.96 |
| 25$\times$4 | 1.02 | 2.40 | 3.42 | 8.25 | 16.44 | 84.08 | 180.89 |
| 200$\times$1 | 0.37 | 0.68 | 1.12 | 3.48 | 6.76 | 33.95 | 70.59 |
| 50$\times$4 | 0.74 | 1.44 | 2.33 | 4.49 | 8.33 | 48.20 | 103.87 |

**Distributed-memory** We estimate a precision matrix with one million dimensions ($p = 10^6$), which contains one trillion parameters ($p^2 = 10^{12}$). The experiments are run on a cluster with 400 computing nodes. We use 1 core per node to avoid the competition for the resources as we observed in the shared-memory case. For $q$ cores, we use the process grid $\frac{q}{2} \times 2$ since $p \gg n$. The block size $p_b \times n_b \times k_b$ for matrix multiplication is $10 \times 10 \times 1$ for $k \leq 10$ and $10 \times 10 \times 10$ for $k > 10$. Since the column block CLIME problems are totally independent, we report the speedups on solving a single column block. The speedup $S_k^{col}$ is plotted in Figure 4(a), where the speedups are larger and more stable than that in the shared-memory environment. The speedup keeps increasing before arriving at a certain number as column block size increases. For any column block size, the speedup also increases as the number of cores increases. The speedup $S_q^{core}$ is plotted in Figure 4(b), where $T_1^{core}$ is the time on 50 cores. A single column ($k = 1$) fails to achieve linear speedups when hundreds of cores are used. However, if using a column block $k > 1$, the ideal linear speedups are achieved with increasing number of cores. Note that due to distributed memory, the larger column block sizes also scale linearly, unlike in the shared memory setting, where the speedups were limited due to resource sharing. As we have seen, $k$ depends on the size of process grid, block size in matrix multiplication, cache size and probably the sparsity pattern of matrices. In Table 2, we compare the performance of 1 core per node to that of using 4 cores per node, which mixes the effects of shared-memory and distributed-memory architectures. For small column block size ($k = 1, 5$), the use of multiple cores in a node is almost two times slower than the use of a single core in a node. For other column block sizes, it is still 30% slower. Finally, we ran CLIME-ADMM on 400 cores with one node per core and block size $k = 500$, and the entire computation took about 11 hours.

## 6 Conclusions

In this paper, we presented a large scale distributed framework for the estimation of sparse precision matrix using CLIME. Our framework can scale to millions of dimensions and run on hundreds of machines. The framework is based on inexact ADMM, which decomposes the constrained optimization problem into elementary matrix multiplications and elementwise operations. Convergence rates for both the objective and optimality conditions are established. The proposed framework solves the CLIME in column-blocks and uses block cyclic distribution to achieve load balancing. We evaluate our algorithm on both shared-memory and distributed-memory architectures. Experimental results show that our algorithm is substantially more scalable than state-of-the-art methods and scales almost linearly with the number of cores. The framework presented can be useful for a variety of other large scale constrained optimization problems and will be explored in future work.

## Acknowledgment

H. W. and A. B. acknowledge the support of NSF via IIS-0953274, IIS-1029711, IIS- 0916750, IIS-0812183, and the technical support from the University of Minnesota Supercomputing Institute. H. W. acknowledges the support of DDF (2013-2014) from the University of Minnesota. C.-J.H. and I.S.D was supported by NSF grants CCF-1320746 and CCF-1117055. C.-J.H also acknowledge the support of IBM PhD fellowship. P.R. acknowledges the support of NSF via IIS-1149803, DMS-1264033 and ARO via W911NF-12-1-0390.

## Footnotes

[1]The interior point method in [3] is written in R and extremely slow. Therefore, we use flare which is implemented in C with R interface. http://cran.r-project.org/web/packages/flare/index.html

[2]www.esrl.noaa.gov/psd/data/gridded/data.ncep.reanalysis.surface.html

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
