[Supplementary Material]

# Supplement:
# Large Scale Distributed Sparse Precision Estimation

**Huahua Wang, Arindam Banerjee**
Dept. of Computer Science & Engg, University of Minnesota, Twin Cities
{huwang,banerjee}@cs.umn.edu

**Cho-Jui Hsieh, Pradeep Ravikumar, Inderjit S. Dhillon**
Dept. of Computer Science, University of Texas, Austin
{cjhsieh,pradeepr,inderjit}@cs.utexas.edu

## A  Optimization Convergence Rate for CLIME ADMM

All norms in this section are defined elementwise. To recap, we solve the following problem:

$$\min \|\mathbf{X}\|_1 \quad \text{s.t.} \quad \|\mathbf{Z} - \mathbf{E}\|_\infty \le \lambda, \mathbf{CX} = \mathbf{Z} \,. \tag{1}$$

The Lagrangian of (1) is

$$L(\mathbf{X}, \mathbf{Z}, \mathbf{Y}) = \|\mathbf{X}\|_1 + \rho\langle \mathbf{Y}, \mathbf{CX} - \mathbf{Z}\rangle \,, \tag{2}$$

where $\|\mathbf{Z} - \mathbf{E}\|_\infty \le \lambda$. Assume that $\{\mathbf{X}^*, \mathbf{Z}^*, \mathbf{Y}^*\}$ satisfies the KKT conditions of (2), i.e.,

$$-\rho\mathbf{C}^T\mathbf{Y}^* \in \partial\|\mathbf{X}^*\|_1 \,, \tag{3}$$

$$\langle \mathbf{Y}^*, \mathbf{Z}^* - \mathbf{Z}\rangle \ge 0 \,, \tag{4}$$

$$\mathbf{CX}^* = \mathbf{Z}^* \,. \tag{5}$$

where (4) holds for any $\mathbf{Z}$ satisfying $\|\mathbf{Z} - \mathbf{E}\|_\infty \le \lambda$. $\{\mathbf{X}^*, \mathbf{Z}^*, \mathbf{Y}^*\}$ is an optimal solution, which has the following property.

**Lemma 1** *Let* $\{\mathbf{X}^t, \mathbf{Z}^t, \mathbf{Y}^t\}$ *be generated by ADMM and* $\{\mathbf{X}^*, \mathbf{Z}^*, \mathbf{Y}^*\}$ *be a KKT point. We have*

$$\|\mathbf{X}^*\|_1 - \|\mathbf{X}^{t+1}\|_1 \le \rho\langle \mathbf{Y}^*, \mathbf{CX}^{t+1} - \mathbf{Z}^{t+1}\rangle \,. \tag{6}$$

*Proof:*  Assume $\{\mathbf{X}^*, \mathbf{Z}^*, \mathbf{Y}^*\}$ is a KKT point. Using the convexity of $\ell_1$ norm and (3), we have

$$\|\mathbf{X}^*\|_1 - \|\mathbf{X}^{t+1}\|_1 \le -\rho\langle \mathbf{CY}^*, \mathbf{X}^* - \mathbf{X}^{t+1}\rangle = -\rho\langle \mathbf{Y}^*, \mathbf{C}(\mathbf{X}^* - \mathbf{X}^{t+1})\rangle \,. \tag{7}$$

Setting $\mathbf{Z} = \mathbf{Z}^{t+1}$ in (4) yields

$$0 \le \langle \mathbf{Y}^*, \mathbf{Z}^* - \mathbf{Z}^{t+1}\rangle \,. \tag{8}$$

Multiplying by $\rho$ and adding to (7) complete the proof.  ∎

In CLIME ADMM, we have the following iterates:

$$\mathbf{X}^{t+1} = \operatorname{argmin}_{\mathbf{X}} \|\mathbf{X}\|_1 + \eta\langle \mathbf{V}^t, \mathbf{X}\rangle + \frac{\eta}{2}\|\mathbf{X} - \mathbf{X}^t\|_2^2 \,, \tag{9}$$

$$\mathbf{Z}^{t+1} = \operatorname*{argmin}_{\|\mathbf{Z}-\mathbf{E}\|_\infty \le \lambda} \frac{\rho}{2}\|\mathbf{CX}^{t+1} - \mathbf{Z} + \mathbf{Y}^t\|_2^2 \,, \tag{10}$$

$$\mathbf{Y}^{t+1} = \mathbf{Y}^t + \mathbf{CX}^{t+1} - \mathbf{Z}^{t+1} \,. \tag{11}$$

where $\mathbf{V}^t = \frac{\rho}{\eta}\mathbf{C}(\mathbf{Y}^t + \mathbf{CX}^t - \mathbf{Z}^t)$.

Throughout the proof of convergence rate, we need the following lemma.

**Lemma 2** *Let* $\mathbf{A}, \mathbf{B}, \mathbf{C}, \mathbf{D}$ *be matrices of the same size. The following equalities hold:*

$$\langle \mathbf{A} - \mathbf{B}, \mathbf{B} - \mathbf{C} \rangle = \frac{1}{2}(\|\mathbf{A} - \mathbf{C}\|_2^2 - \|\mathbf{A} - \mathbf{B}\|_2^2 - \|\mathbf{B} - \mathbf{C}\|_2^2) . \tag{12}$$

$$\langle \mathbf{A} - \mathbf{B}, \mathbf{C} - \mathbf{D} \rangle = \frac{1}{2}(\|\mathbf{D} - \mathbf{A}\|_2^2 - \|\mathbf{D} - \mathbf{B}\|_2^2 + \|\mathbf{C} - \mathbf{B}\|_2^2 - \|\mathbf{C} - \mathbf{A}\|_2^2) . \tag{13}$$

### A.0.1 $O(1/T)$ Convergence Rate for Objective Function

In this section, we establish the iteration complexity for inexact ADMM (9)-(11). We begin with the following lemma for the $\mathbf{X}$ update (9).

**Lemma 3** *Let* $\{\mathbf{X}^t, \mathbf{Z}^t, \mathbf{Y}^t\}$ *be generated by (9)-(11). For any* $\mathbf{X}$*, we have*

$$\|\mathbf{X}^{t+1}\|_1 - \|\mathbf{X}\|_1 \leq -\rho \langle \mathbf{Y}^{t+1}, \mathbf{C}(\mathbf{X}^{t+1} - \mathbf{X}) \rangle + \frac{\rho}{2}(\|\mathbf{C}\mathbf{X} - \mathbf{Z}^t\|_2^2 - \|\mathbf{C}\mathbf{X} - \mathbf{Z}^{t+1}\|_2^2 + \|\mathbf{C}\mathbf{X}^{t+1} - \mathbf{Z}^{t+1}\|_2^2$$

$$- \|\mathbf{C}\mathbf{X}^{t+1} - \mathbf{Z}^t\|_2^2) + \frac{1}{2}(\|\mathbf{X} - \mathbf{X}^t\|_{\eta\mathbf{I} - \rho\mathbf{C}^2}^2 - \|\mathbf{X} - \mathbf{X}^{t+1}\|_{\eta\mathbf{I} - \rho\mathbf{C}^2}^2 - \|\mathbf{X}^{t+1} - \mathbf{X}^t\|_{\eta\mathbf{I} - \rho\mathbf{C}^2}^2) . \tag{14}$$

*Proof:* Let $\partial\|\mathbf{X}^{t+1}\|_1$ be the subgradient of $\|\mathbf{X}^{t+1}\|_1$. Since $\mathbf{X}^{t+1}$ is a minimizer of (9), we have

$$\mathbf{0} \in \partial\|\mathbf{X}^{t+1}\|_1 + \eta(\mathbf{V}^t + \mathbf{X}^{t+1} - \mathbf{X}^t) . \tag{15}$$

Rearranging the terms gives $-\eta(\mathbf{V}^t + \mathbf{X}^{t+1} - \mathbf{X}^t) \in \partial\|\mathbf{X}^{t+1}\|_1$. Using the convexity of $\ell_1$ norm, we have

$$\|\mathbf{X}^{t+1}\|_1 - \|\mathbf{X}\|_1 \leq -\eta \langle \mathbf{V}^t + \mathbf{X}^{t+1} - \mathbf{X}^t, \mathbf{X}^{t+1} - \mathbf{X} \rangle$$

$$\leq -\rho \langle \mathbf{C}(\mathbf{Y}^t + \mathbf{C}\mathbf{X}^t - \mathbf{Z}^t), \mathbf{X}^{t+1} - \mathbf{X} \rangle - \eta \langle \mathbf{X}^{t+1} - \mathbf{X}^t, \mathbf{X}^{t+1} - \mathbf{X} \rangle \tag{16}$$

$$\leq -\rho \langle \mathbf{Y}^t + \mathbf{C}\mathbf{X}^t - \mathbf{Z}^t, \mathbf{C}(\mathbf{X}^{t+1} - \mathbf{X}) \rangle - \eta \langle \mathbf{X}^{t+1} - \mathbf{X}^t, \mathbf{X}^{t+1} - \mathbf{X} \rangle$$

$$= -\rho \langle \mathbf{Y}^{t+1}, \mathbf{C}(\mathbf{X}^{t+1} - \mathbf{X}) \rangle - \rho \langle \mathbf{C}(\mathbf{X}^t - \mathbf{X}^{t+1}), \mathbf{C}(\mathbf{X}^{t+1} - \mathbf{X}) \rangle + \rho \langle \mathbf{Z}^t - \mathbf{Z}^{t+1}, \mathbf{C}(\mathbf{X}^{t+1} - \mathbf{X}) \rangle$$

$$- \eta \langle \mathbf{X}^{t+1} - \mathbf{X}^t, \mathbf{X}^{t+1} - \mathbf{X} \rangle . \tag{17}$$

where the last equality uses (11). Using (12), the second term can be written as

$$-\langle \mathbf{C}(\mathbf{X}^t - \mathbf{X}^{t+1}), \mathbf{C}(\mathbf{X}^{t+1} - \mathbf{X}) \rangle = -\frac{1}{2}(\|\mathbf{C}(\mathbf{X} - \mathbf{X}^t)\|_2^2 - \|\mathbf{C}(\mathbf{X} - \mathbf{X}^{t+1})\|_2^2 - \|\mathbf{C}(\mathbf{X}^t - \mathbf{X}^{t+1})\|_2^2) . \tag{18}$$

Note $\|\mathbf{C}(\mathbf{X} - \mathbf{X}^t)\|_2^2 = \|\mathbf{X} - \mathbf{X}^t\|_{\mathbf{C}^2}^2$. Using (13), the third term of (17) can be written as

$$\langle \mathbf{Z}^t - \mathbf{Z}^{t+1}, \mathbf{C}(\mathbf{X}^{t+1} - \mathbf{X}) \rangle = \frac{1}{2}(\|\mathbf{C}\mathbf{X} - \mathbf{Z}^t\|_2^2 - \|\mathbf{C}\mathbf{X} - \mathbf{Z}^{t+1}\|_2^2 + \|\mathbf{C}\mathbf{X}^{t+1} - \mathbf{Z}^{t+1}\|_2^2 - \|\mathbf{C}\mathbf{X}^{t+1} - \mathbf{Z}^t\|_2^2) . \tag{19}$$

Applying (12) on the last term of (17) gives

$$-\langle \mathbf{X}^{t+1} - \mathbf{X}^t, \mathbf{X}^{t+1} - \mathbf{X} \rangle = \frac{1}{2}(\|\mathbf{X} - \mathbf{X}^t\|_2^2 - \|\mathbf{X} - \mathbf{X}^{t+1}\|_2^2 - \|\mathbf{X}^{t+1} - \mathbf{X}^t\|_2^2) . \tag{20}$$

Substituting (18)-(20) into (17) and rearraning the terms complete the proof. ∎

The $\mathbf{Z}$ update (10) has the following lemma.

**Lemma 4** *Let* $\{\mathbf{X}^t, \mathbf{Z}^t, \mathbf{Y}^t\}$ *be generated by (9)-(11). For any* $\mathbf{Z}$ *satisfying* $\|\mathbf{Z} - \mathbf{E}\|_\infty \leq \lambda$,

$$0 \leq -\langle \mathbf{Y}^{t+1}, \mathbf{Z} - \mathbf{Z}^{t+1} \rangle . \tag{21}$$

*Proof:* Since $\mathbf{Z}^{t+1}$ is a minimizer of (10), for any $\mathbf{Z}$ satisfying the infinity norm constraint, then

$$-\langle \mathbf{C}\mathbf{X}^{t+1} - \mathbf{Z}^{t+1} + \mathbf{Y}^t, \mathbf{Z} - \mathbf{Z}^{t+1} \rangle \geq 0 . \tag{22}$$

Using (11) completes the proof. ∎

Combining the results in Lemma 3 and 4 yields the $O(1/T)$ convergence rate for the objective of inexact ADMM (9)-(11).

**Theorem 1** *Let* $\{\mathbf{X}^t, \mathbf{Z}^t, \mathbf{Y}^t\}$ *be generated by (10)-(11) and* $\bar{\mathbf{X}}^T = \frac{1}{T}\sum_{t=1}^{T} \mathbf{X}^t$. *Assume* $\mathbf{X}^0 = \mathbf{Z}^0 = \mathbf{Y}^0 = \mathbf{0}$ *and* $\eta \geq \lambda_{\max}^2(\mathbf{C})$. *For any* $\mathbf{CX} = \mathbf{Z}$, *we have*

$$\|\bar{\mathbf{X}}^T\|_1 - \|\mathbf{X}\|_1 \leq \frac{\eta\|\mathbf{X}\|_2^2}{2T} \ . \tag{23}$$

*Proof:* Assume $\mathbf{CX} = \mathbf{Z}$. Multiplying (21) by $\rho$ and adding (14) yields

$$\|\mathbf{X}^{t+1}\|_1 - \|\mathbf{X}\|_1 \leq -\rho\langle \mathbf{Y}^{t+1}, \mathbf{CX}^{t+1} - \mathbf{Z}^{t+1}\rangle + \frac{1}{2}(\|\mathbf{Z} - \mathbf{Z}^t\|_2^2 - \|\mathbf{Z} - \mathbf{Z}^{t+1}\|_2^2 + \|\mathbf{CX}^{t+1} - \mathbf{Z}^{t+1}\|_2^2$$

$$- \|\mathbf{CX}^{t+1} - \mathbf{Z}^t\|_2^2) + \frac{1}{2}(\|\mathbf{X} - \mathbf{X}^t\|_{\eta\mathbf{I}-\rho\mathbf{C}^2}^2 - \|\mathbf{X} - \mathbf{X}^{t+1}\|_{\eta\mathbf{I}-\rho\mathbf{C}^2}^2 - \|\mathbf{X}^{t+1} - \mathbf{X}^t\|_{\eta\mathbf{I}-\rho\mathbf{C}^2}^2) \ . \tag{24}$$

Using (11), the first term can be written as

$$-\langle \mathbf{Y}^{t+1}, \mathbf{CX}^{t+1} - \mathbf{Z}^{t+1}\rangle = -\langle \mathbf{Y}^{t+1}, \mathbf{Y}^{t+1} - \mathbf{Y}^t\rangle$$

$$= \frac{1}{2}(\|\mathbf{Y}^t\|_2^2 - \|\mathbf{Y}^{t+1}\|_2^2 - \|\mathbf{Y}^{t+1} - \mathbf{Y}^t\|_2^2)$$

$$= \frac{1}{2}(\|\mathbf{Y}^t\|_2^2 - \|\mathbf{Y}^{t+1}\|_2^2 - \|\mathbf{CX}^{t+1} - \mathbf{Z}^{t+1}\|_2^2) \ . \tag{25}$$

Substituting back into (24) gives

$$\|\mathbf{X}^{t+1}\|_1 - \|\mathbf{X}\|_1 \leq \frac{\rho}{2}(\|\mathbf{Y}^t\|_2^2 - \|\mathbf{Y}^{t+1}\|_2^2) + \frac{\rho}{2}(\|\mathbf{Z} - \mathbf{Z}^t\|_2^2 - \|\mathbf{Z} - \mathbf{Z}^{t+1}\|_2^2 - \|\mathbf{CX}^{t+1} - \mathbf{Z}^t\|_2^2)$$

$$+ \frac{1}{2}(\|\mathbf{X} - \mathbf{X}^t\|_{\eta\mathbf{I}-\rho\mathbf{C}^2}^2 - \|\mathbf{X} - \mathbf{X}^{t+1}\|_{\eta\mathbf{I}-\rho\mathbf{C}^2}^2 - \|\mathbf{X}^{t+1} - \mathbf{X}^t\|_{\eta\mathbf{I}-\rho\mathbf{C}^2}^2) \ . \tag{26}$$

Assuming $\eta \geq \lambda_{\max}^2(\mathbf{C})$, $\eta\mathbf{I} - \rho\mathbf{C}^2$ is positive semidefinite. Summing over $t$ from $0$ to $T-1$ and ignoring some negative terms, we have the following telescoping sum

$$\sum_{t=0}^{T-1} \|\mathbf{X}^{t+1}\|_1 - \|\mathbf{X}\|_1 \leq \frac{\rho}{2}\|\mathbf{Y}^0\|_2^2 + \frac{\rho}{2}\|\mathbf{Z} - \mathbf{Z}^0\|_2^2 + \frac{1}{2}\|\mathbf{X} - \mathbf{X}^0\|_{\eta\mathbf{I}-\rho\mathbf{C}^2}^2$$

$$= \frac{\rho}{2}\|\mathbf{Z}\|_2^2 + \frac{1}{2}\|\mathbf{X}\|_{\eta\mathbf{I}-\rho\mathbf{C}^2}^2$$

$$= \frac{\eta}{2}\|\mathbf{X}\|_2^2 \ . \tag{27}$$

where the first equality is due to $\mathbf{X}^0 = \mathbf{Z}^0 = \mathbf{Y}^0 = \mathbf{0}$ and the second equality uses $\mathbf{CX} = \mathbf{Z}$. Applying the Jensen's inequality on the left hand side completes the proof. ∎

### A.0.2 $O(1/T)$ **Convergence Rate for the Optimality Conditions**

For the $\mathbf{X}$ update (9), we have the following lemma.

**Lemma 5** *Let* $\{\mathbf{X}^t, \mathbf{Z}^t, \mathbf{Y}^t\}$ *be generated by (9)-(11). We have*

$$\|\mathbf{CX}^{t+1} - \mathbf{Z}^t\|_2^2 + \|\mathbf{X}^{t+1} - \mathbf{X}^t\|_{\frac{\eta}{\rho}\mathbf{I}-\mathbf{C}^2}^2 \leq \|\mathbf{CX}^t - \mathbf{Z}^t\|_2^2 + \|\mathbf{Z}^{t-1} - \mathbf{Z}^t\|_2^2 + \|\mathbf{X}^t - \mathbf{X}^{t-1}\|_{\frac{\eta}{\rho}\mathbf{I}-\mathbf{C}^2}^2 \ . \tag{28}$$

*Proof:* Setting $\mathbf{X} = \mathbf{X}^t$ in (16) gives

$$\|\mathbf{X}^{t+1}\|_1 - \|\mathbf{X}^t\|_1 \leq -\rho\langle \mathbf{Y}^t + \mathbf{CX}^t - \mathbf{Z}^t, \mathbf{C}(\mathbf{X}^{t+1} - \mathbf{X}^t)\rangle - \eta\langle \mathbf{X}^{t+1} - \mathbf{X}^t, \mathbf{X}^{t+1} - \mathbf{X}^t\rangle$$

$$\leq -\rho\langle \mathbf{Y}^t, \mathbf{C}(\mathbf{X}^{t+1} - \mathbf{X}^t)\rangle + \frac{\rho}{2}(\|\mathbf{CX}^t - \mathbf{Z}^t\|_2^2 + \|\mathbf{C}(\mathbf{X}^{t+1} - \mathbf{X}^t)\|_2^2 - \|\mathbf{CX}^{t+1} - \mathbf{Z}^t\|_2^2) - \eta\|\mathbf{X}^{t+1} - \mathbf{X}^t\|_2^2 \ . \tag{29}$$

At $t$, (17) becomes

$$\|\mathbf{X}^t\|_1 - \|\mathbf{X}\|_1 \leq -\rho\langle \mathbf{Y}^t, \mathbf{C}(\mathbf{X}^t - \mathbf{X})\rangle - \rho\langle \mathbf{C}(\mathbf{X}^{t-1} - \mathbf{X}^t), \mathbf{C}(\mathbf{X}^t - \mathbf{X})\rangle$$

$$+ \rho\langle \mathbf{Z}^{t-1} - \mathbf{Z}^t, \mathbf{C}(\mathbf{X}^t - \mathbf{X})\rangle - \eta\langle \mathbf{X}^t - \mathbf{X}^{t-1}, \mathbf{X}^t - \mathbf{X}\rangle \ . \tag{30}$$

Setting $\mathbf{X} = \mathbf{X}^{t+1}$ gives

$$\|\mathbf{X}^t\|_1 - \|\mathbf{X}^{t+1}\|_1 \leq -\rho\langle \mathbf{Y}^t, \mathbf{C}(\mathbf{X}^t - \mathbf{X}^{t+1})\rangle - \rho\langle \mathbf{C}(\mathbf{X}^{t-1} - \mathbf{X}^t), \mathbf{C}(\mathbf{X}^t - \mathbf{X}^{t+1})\rangle$$
$$+ \rho\langle \mathbf{Z}^{t-1} - \mathbf{Z}^t, \mathbf{C}(\mathbf{X}^t - \mathbf{X}^{t+1})\rangle - \eta\langle \mathbf{X}^t - \mathbf{X}^{t-1}, \mathbf{X}^t - \mathbf{X}^{t+1}\rangle. \quad (31)$$

Using (12), the second term becomes

$$-\rho\langle \mathbf{C}(\mathbf{X}^{t-1} - \mathbf{X}^t), \mathbf{C}(\mathbf{X}^t - \mathbf{X}^{t+1})\rangle$$
$$= -\frac{\rho}{2}(\|\mathbf{C}(\mathbf{X}^{t-1} - \mathbf{X}^{t+1})\|_2^2 - \|\mathbf{C}(\mathbf{X}^{t-1} - \mathbf{X}^t)\|_2^2 - \|\mathbf{C}(\mathbf{X}^t - \mathbf{X}^{t+1})\|_2^2). \quad (32)$$

Similarly, applying (12) on the fourth term of (31) gives

$$-\eta\langle \mathbf{X}^t - \mathbf{X}^{t-1}, \mathbf{X}^t - \mathbf{X}^{t+1}\rangle = \frac{\eta}{2}(\|\mathbf{X}^{t-1} - \mathbf{X}^{t+1}\|_2^2 - \|\mathbf{X}^t - \mathbf{X}^{t-1}\|_2^2 - \|\mathbf{X}^t - \mathbf{X}^{t+1}\|_2^2). \quad (33)$$

Adding (32) and (33) together yields

$$-\rho\langle \mathbf{C}(\mathbf{X}^{t-1} - \mathbf{X}^t), \mathbf{C}(\mathbf{X}^t - \mathbf{X}^{t+1})\rangle - \eta\langle \mathbf{X}^t - \mathbf{X}^{t-1}, \mathbf{X}^t - \mathbf{X}^{t+1}\rangle$$
$$= \frac{1}{2}(\|\mathbf{X}^{t-1} - \mathbf{X}^{t+1}\|_{\eta\mathbf{I}-\rho\mathbf{C}^2}^2 - \|\mathbf{X}^t - \mathbf{X}^{t-1}\|_{\eta\mathbf{I}-\rho\mathbf{C}^2}^2 - \|\mathbf{X}^t - \mathbf{X}^{t+1}\|_{\eta\mathbf{I}-\rho\mathbf{C}^2}^2)$$
$$\leq \frac{1}{2}(\|\mathbf{X}^t - \mathbf{X}^{t-1}\|_{\eta\mathbf{I}-\rho\mathbf{C}^2}^2 + \|\mathbf{X}^t - \mathbf{X}^{t+1}\|_{\eta\mathbf{I}-\rho\mathbf{C}^2}^2), \quad (34)$$

where the last inequality uses $\|\mathbf{A} - \mathbf{B}\|_2^2 \leq 2(\|\mathbf{A} - \mathbf{C}\|_2^2 + \|\mathbf{B} - \mathbf{C}\|_2^2)$. Using the inequality $\langle \mathbf{A}, \mathbf{B}\rangle \leq \frac{1}{2}(\|\mathbf{A}\|_2^2 + \|\mathbf{B}\|_2^2)$, the third term of (31) can be written as

$$\rho\langle \mathbf{Z}^{t-1} - \mathbf{Z}^t, \mathbf{C}(\mathbf{X}^t - \mathbf{X}^{t+1})\rangle \leq \frac{\rho}{2}(\|\mathbf{Z}^{t-1} - \mathbf{Z}^t\|_2^2 + \|\mathbf{C}(\mathbf{X}^t - \mathbf{X}^{t+1})\|_2^2). \quad (35)$$

Substituting (34) and (35) back to (31), we have

$$\|\mathbf{X}^t\|_1 - \|\mathbf{X}^{t+1}\|_1 \leq -\rho\langle \mathbf{Y}^t, \mathbf{C}(\mathbf{X}^t - \mathbf{X}^{t+1})\rangle + \frac{\rho}{2}\|\mathbf{Z}^{t-1} - \mathbf{Z}^t\|_2^2$$
$$+ \frac{1}{2}(\|\mathbf{X}^t - \mathbf{X}^{t-1}\|_{\eta\mathbf{I}-\rho\mathbf{C}^2}^2 + \eta\|\mathbf{X}^t - \mathbf{X}^{t+1}\|_2^2) \quad (36)$$

Adding (29) and (36) together yields

$$0 \leq \frac{\rho}{2}(\|\mathbf{C}\mathbf{X}^t - \mathbf{Z}^t\|_2^2 + \|\mathbf{C}(\mathbf{X}^{t+1} - \mathbf{X}^t)\|_2^2 - \|\mathbf{C}\mathbf{X}^{t+1} - \mathbf{Z}^t\|_2^2) - \eta\|\mathbf{X}^{t+1} - \mathbf{X}^t\|_2^2$$
$$+ \frac{\rho}{2}\|\mathbf{Z}^{t-1} - \mathbf{Z}^t\|_2^2 + \frac{1}{2}(\|\mathbf{X}^t - \mathbf{X}^{t-1}\|_{\eta\mathbf{I}-\rho\mathbf{C}^2}^2 + \eta\|\mathbf{X}^t - \mathbf{X}^{t+1}\|_2^2)$$
$$= \frac{\rho}{2}(\|\mathbf{C}\mathbf{X}^t - \mathbf{Z}^t\|_2^2 + \|\mathbf{Z}^{t-1} - \mathbf{Z}^t\|_2^2 - \|\mathbf{C}\mathbf{X}^{t+1} - \mathbf{Z}^t\|_2^2)$$
$$+ \frac{1}{2}(\|\mathbf{X}^t - \mathbf{X}^{t-1}\|_{\eta\mathbf{I}-\rho\mathbf{C}^2}^2 - \|\mathbf{X}^{t+1} - \mathbf{X}^t\|_{\eta\mathbf{I}-\rho\mathbf{C}^2}^2). \quad (37)$$

Dividing both sides by $\frac{\rho}{2}$ and rearranging the terms complete the proof. ∎

For the $\mathbf{Z}$ update (10), we have the following lemma.

**Lemma 6** *Let $\{\mathbf{X}^t, \mathbf{Z}^t, \mathbf{Y}^t\}$ be generated by (9)-(11). We have*

$$\|\mathbf{C}\mathbf{X}^{t+1} - \mathbf{Z}^{t+1}\|_2^2 + \|\mathbf{Z}^{t+1} - \mathbf{Z}^t\|_2^2 \leq \|\mathbf{C}\mathbf{X}^{t+1} - \mathbf{Z}^t\|_2^2. \quad (38)$$

*Proof:* Setting $\mathbf{Z} = \mathbf{Z}^t$ in (21) gives

$$0 \leq -\langle \mathbf{Y}^{t+1}, \mathbf{Z}^t - \mathbf{Z}^{t+1}\rangle. \quad (39)$$

At $t$, (21) becomes

$$0 \leq -\langle \mathbf{Y}^t, \mathbf{Z} - \mathbf{Z}^t\rangle. \quad (40)$$

Setting $\mathbf{Z} = \mathbf{Z}^{t+1}$ yields

$$0 \leq -\langle \mathbf{Y}^t, \mathbf{Z}^{t+1} - \mathbf{Z}^t \rangle . \tag{41}$$

Adding (39) and (41) yields

$$0 \leq \langle \mathbf{Y}^{t+1} - \mathbf{Y}^t, \mathbf{Z}^{t+1} - \mathbf{Z}^t \rangle = \langle \mathbf{CX}^{t+1} - \mathbf{Z}^{t+1}, \mathbf{Z}^{t+1} - \mathbf{Z}^t \rangle$$
$$= \frac{1}{2}(\|\mathbf{CX}^{t+1} - \mathbf{Z}^t\|_2^2 - \|\mathbf{CX}^{t+1} - \mathbf{Z}^{t+1}\|_2^2 - \|\mathbf{Z}^{t+1} - \mathbf{Z}^t\|_2^2) . \tag{42}$$

Rearranging the terms complete the proof. ∎

Define $R_1(t+1)$ as follows:

$$R_1(t+1) = \|\mathbf{CX}^{t+1} - \mathbf{Z}^{t+1}\|_2^2 + \|\mathbf{Z}^{t+1} - \mathbf{Z}^t\|_2^2 + \|\mathbf{X}^{t+1} - \mathbf{X}^t\|_{\frac{\eta}{\rho}\mathbf{I} - \mathbf{C}^2}^2 . \tag{43}$$

We now show that $R_1(t)$ is non-increasing by combining the results in Lemma 5 and 6 .

**Lemma 7** *Let $R_1(t)$ be defined in (43). We have*

$$R_1(t+1) \leq R_1(t) . \tag{44}$$

*Proof:* Adding (28) and (38) yields

$$\|\mathbf{CX}^{t+1} - \mathbf{Z}^{t+1}\|_2^2 + \|\mathbf{Z}^{t+1} - \mathbf{Z}^t\|_2^2 + \|\mathbf{X}^{t+1} - \mathbf{X}^t\|_{\frac{\eta}{\rho}\mathbf{I} - \mathbf{C}^2}^2$$
$$\leq \|\mathbf{CX}^t - \mathbf{Z}^t\|_2^2 + \|\mathbf{Z}^{t-1} - \mathbf{Z}^t\|_2^2 + \|\mathbf{X}^t - \mathbf{X}^{t-1}\|_{\frac{\eta}{\rho}\mathbf{I} - \mathbf{C}^2}^2 . \tag{45}$$

(44) follows from the definition of $R_1$ in (43). ∎

**Lemma 8** *Let $\{\mathbf{X}^t, \mathbf{Z}^t, \mathbf{Y}^t\}$ be generated by (9)-(11) and $\{\mathbf{X}^*, \mathbf{Z}^*, \mathbf{Y}^*\}$ be a KKT point. We have*

$$R_1(t+1) \leq \|\mathbf{Y}^* - \mathbf{Y}^t\|_2^2 - \|\mathbf{Y}^* - \mathbf{Y}^{t+1}\|_2^2 + \|\mathbf{Z}^* - \mathbf{Z}^t\|_2^2 - \|\mathbf{Z}^* - \mathbf{Z}^{t+1}\|_2^2$$
$$+ \|\mathbf{X}^* - \mathbf{X}^t\|_{\frac{\eta}{\rho}\mathbf{I} - \mathbf{C}^2}^2 - \|\mathbf{X}^* - \mathbf{X}^{t+1}\|_{\frac{\eta}{\rho}\mathbf{I} - \mathbf{C}^2}^2 . \tag{46}$$

*where $R_1(t+1)$ is defined in (43).*

*Proof:* Adding (24) and (6) yields

$$0 \leq \rho\langle \mathbf{Y}^* - \mathbf{Y}^{t+1}, \mathbf{CX}^{t+1} - \mathbf{Z}^{t+1} \rangle + \frac{\rho}{2}(\|\mathbf{Z}^* - \mathbf{Z}^t\|_2^2 - \|\mathbf{Z}^* - \mathbf{Z}^{t+1}\|_2^2 + \|\mathbf{CX}^{t+1} - \mathbf{Z}^{t+1}\|_2^2$$
$$- \|\mathbf{CX}^{t+1} - \mathbf{Z}^t\|_2^2) + \frac{1}{2}(\|\mathbf{X}^* - \mathbf{X}^t\|_{\eta\mathbf{I} - \rho\mathbf{C}^2}^2 - \|\mathbf{X}^* - \mathbf{X}^{t+1}\|_{\eta\mathbf{I} - \rho\mathbf{C}^2}^2 - \|\mathbf{X}^{t+1} - \mathbf{X}^t\|_{\eta\mathbf{I} - \rho\mathbf{C}^2}^2) . \tag{47}$$

Using (11) and applying (12) on the first term, we have

$$\langle \mathbf{Y}^* - \mathbf{Y}^{t+1}, \mathbf{CX}^{t+1} - \mathbf{Z}^{t+1} \rangle = \langle \mathbf{Y}^* - \mathbf{Y}^{t+1}, \mathbf{Y}^{t+1} - \mathbf{Y}^t \rangle$$
$$= \frac{1}{2}(\|\mathbf{Y}^* - \mathbf{Y}^t\|_2^2 - \|\mathbf{Y}^* - \mathbf{Y}^{t+1}\|_2^2 - \|\mathbf{Y}^{t+1} - \mathbf{Y}^t\|_2^2)$$
$$= \frac{1}{2}(\|\mathbf{Y}^* - \mathbf{Y}^t\|_2^2 - \|\mathbf{Y}^* - \mathbf{Y}^{t+1}\|_2^2 - \|\mathbf{CX}^{t+1} - \mathbf{Z}^{t+1}\|_2^2) . \tag{48}$$

Plugging into (47) yields

$$0 \leq \frac{\rho}{2}(\|\mathbf{Y}^* - \mathbf{Y}^t\|_2^2 - \|\mathbf{Y}^* - \mathbf{Y}^{t+1}\|_2^2) + \frac{\rho}{2}(\|\mathbf{Z}^* - \mathbf{Z}^t\|_2^2 - \|\mathbf{Z}^* - \mathbf{Z}^{t+1}\|_2^2 - \|\mathbf{CX}^{t+1} - \mathbf{Z}^t\|_2^2)$$
$$+ \frac{1}{2}(\|\mathbf{X}^* - \mathbf{X}^t\|_{\eta\mathbf{I} - \rho\mathbf{C}^2}^2 - \|\mathbf{X}^* - \mathbf{X}^{t+1}\|_{\eta\mathbf{I} - \rho\mathbf{C}^2}^2 - \|\mathbf{X}^{t+1} - \mathbf{X}^t\|_{\eta\mathbf{I} - \rho\mathbf{C}^2}^2) . \tag{49}$$

Dividing both sides by $\frac{\rho}{2}$ and rearraning the terms, we have (46) by using (38) and the definition of $R_1(t)$ in (43). ∎

**Theorem 2** *Let $\{\mathbf{X}^t, \mathbf{Z}^t, \mathbf{Y}^t\}$ be generated by (9)-(11) and $\{\mathbf{X}^*, \mathbf{Z}^*, \mathbf{Y}^*\}$ be a KKT point. Assume $\mathbf{X}^0 = \mathbf{Z}^0 = \mathbf{Y}^0 = \mathbf{0}$ and $\eta \geq \lambda_{\max}^2(\mathbf{C})$. We have*

$$R_1(T) \leq \frac{\|\mathbf{Y}^*\|_2^2 + \frac{\eta}{\rho}\|\mathbf{X}^*\|_2^2}{T} \,, \tag{50}$$

*where $R_1(T)$ is defined in (43).*

*Proof:* Summing (46) over $t$ from $0$ to $T-1$ and ignoring some negative terms yield

$$\sum_{t=0}^{T-1} R_1(t+1) \leq \|\mathbf{Y}^* - \mathbf{Y}^0\|_2^2 + \|\mathbf{Z}^* - \mathbf{Z}^0\|_2^2 + \|\mathbf{X}^* - \mathbf{X}^0\|_{\frac{\eta}{\rho}\mathbf{I}-\mathbf{C}^2}^2$$

$$= \|\mathbf{Y}^*\|_2^2 + \|\mathbf{Z}^*\|_2^2 + \|\mathbf{X}^*\|_{\frac{\eta}{\rho}\mathbf{I}-\mathbf{C}^2}^2$$

$$= \|\mathbf{Y}^*\|_2^2 + \frac{\eta}{\rho}\|\mathbf{X}^*\|_2^2 \,, \tag{51}$$

where the first equality is due to $\mathbf{X}^0 = \mathbf{Z}^0 = \mathbf{Y}^0 = \mathbf{0}$ and the second equality uses $\mathbf{C}\mathbf{X}^* = \mathbf{Z}^*$. According to Lemma 7, $R_1(t)$ is non-increasing. Therefore,

$$T R_1(T) \leq \sum_{t=0}^{T} R_1(t+1) \,. \tag{52}$$

Dividing both sides by $T$ completes the proof. ∎

The optimality condition for (10) is given in Lemma 4, showing that KKT condition (4) is alway satisfied. The optimality conditions for (9) is

$$-\eta(\mathbf{V}^t + \mathbf{X}^{t+1} - \mathbf{X}^t) \in \partial\|\mathbf{X}^{t+1}\|_1 \,. \tag{53}$$

Expanding $\mathbf{C}$ and using (11), it can be rewritten as

$$-\rho\mathbf{C}(\mathbf{Y}^{t+1} + \mathbf{X}^t - \mathbf{X}^{t+1} - \mathbf{Z}^t + \mathbf{Z}^{t+1}) - \eta(\mathbf{X}^{t+1} - \mathbf{X}^t) \in \partial\|\mathbf{X}^{t+1}\|_1 \,. \tag{54}$$

If $\mathbf{X}^{t+1} = \mathbf{X}^t$ and $\mathbf{Z}^{t+1} = \mathbf{Z}^t$, the KKT condition (3) will be satisfied. Therefore, $R_1(T)$ defines the residuals of optimality conditions for (9)-(11). As $R_1(T) \to 0$, $\mathbf{C}\mathbf{X}^T = \mathbf{Z}^T, \mathbf{Z}^T = \mathbf{Z}^{T-1}$ and $\mathbf{X}^T = \mathbf{X}^{T-1}$ and thus the KKT conditions (3)-(5) are satisfied.

## B  Statistical Convergence Rates with Covariance Perturbation

In this section, we analyze the statistical convergence of the CLIME estimator [1] under perturbations of the sample covariance matrix. For the ease of reading, we first define some notations. Let $R_1, \cdots, R_k, \cdots, R_n \in \Re^p$ be $n$ samples generated from a distribution with covariance matrix $\Sigma_0$ and true precision matrix $\Omega_0$. The estimated covariance matrix is denoted as $\hat{\Sigma}$ and the corresponding estimated precision matrix is $\hat{\Omega}$. The pertubed covariance matrix is denoted as $\hat{S}$. The covariance matrix $\mathbf{C}$ in the main text can be either $\hat{\Sigma}$ or $\hat{S}$. The $i$-th element of $R_k$ is denoted as $R_{ik}$. For matrix, we use $ij$ to index the $ij$-th element, e.g., $\hat{\Omega}_{ij}$. $\|\cdot\|_\infty$ and $\|\cdot\|_2$ denote the elementwise norm. $\|\cdot\|_{L_1}$ and $\|\cdot\|_{L_2}$ denote the matrix $L_1$ norm and $L_2$ norm. For the sake of completeness, we start with a brief review of some of the main results for CLIME.

### B.1  CLIME Estimator: Bounds in terms of $\lambda$

For $n$ samples $R_1, \ldots, R_n \in \Re^p$, the sample covariance matrix $\hat{\Sigma}$, is computed as:

$$\hat{\Sigma} = \frac{1}{n}\sum_{k=1}^{n}(R_k - \bar{R})(R_k - \bar{R})^T = \frac{1}{n}\sum_{k=1}^{n}R_k R_k^T - \frac{1}{n}\bar{R}\bar{R}^T \,, \quad \text{where} \quad \bar{R} = \frac{1}{n}\sum_{k=1}^{n}R_k \,. \tag{55}$$

As a result, an entry of the sample covariance matrix is given by:

$$\hat{\Sigma}_{ij} = \frac{1}{n}\sum_{k=1}^{n}R_{ik}R_{jk} - \frac{1}{n}\left(\frac{1}{n}\sum_{k=1}^{n}R_{ik}\right)\left(\frac{1}{n}\sum_{k=1}^{n}R_{jk}\right) \,. \tag{56}$$

The analysis for CLIME [1] considers the following family of precision matrices:

$$\mathcal{U} = \mathcal{U}(M, q, s_0(p)) = \left\{ \Omega : \Omega \succ 0, \|\Omega\|_{L_1} \leq M, \max_{1 \leq i \leq p} \sum_{j=1}^{p} |\Omega_{ij}|^q \leq s_0(p) \right\} , \qquad (57)$$

for $0 \leq q < 1$. Then, the CLIME estimator has the following guarantees:

**Theorem 3** *Let $\Omega_0 \in \mathcal{U}(M, q, s_0(p))$. If $\lambda \geq \|\Omega_0\|_{L_1} \max_{ij} |\hat{\Sigma}_{ij} - \Sigma_{0,ij}|$, then we have*

$$\|\hat{\Omega} - \Omega_0\|_{\infty} \leq 4\|\Omega_0\|_{L_1} \lambda , \qquad (58)$$

$$\|\hat{\Omega} - \Omega_0\|_{L_2} \leq cs_0(p)(4\|\Omega_0\|_{L_1})^{1-q} \lambda^{1-q} , \qquad (59)$$

$$\frac{1}{p}\|\hat{\Omega} - \Omega_0\|_2^2 \leq cs_0(p)(4\|\Omega_0\|_{L_1})^{2-q} \lambda^{2-q} , \qquad (60)$$

*where $c \leq 2(1 + 2^{1-q} + 3^{1-q})$ is a constant.*

Note that the deterministic bounds in Theorem 3 for precision estimation relies on $\|\hat{\Sigma} - \Sigma_0\|_{\infty} = \max_{i,j} |\hat{\Sigma}_{ij} - \Sigma_{0,ij}|$. In the next subsection, we establish tail bounds for the scenario where we (intentionally) perturb each entry of the sample covariance matrix, i.e., we work with $\hat{S}_{ij} = \hat{\Sigma}_{ij} + \Delta_{ij}$ where $\Delta_{ij}$ has a sub-exponential tail.

## B.2  Bounds for $\lambda$

The following two norms will play a role in our analysis: For a scalar random variable $v$, let

$$\|v\|_{\psi_2} = \sup_{p \geq 1} p^{-1/2}(\mathbb{E}|v|^p)^{1/p} , \quad \text{and} \quad \|v\|_{\psi_1} = \sup_{p \geq 1} p^{-1}(\mathbb{E}|v|^p)^{1/p} . \qquad (61)$$

Then, $v$ is called a *sub-Gaussian* random variable if $\|v\|_{\psi_2} \leq K_2$ for a constant $K_2$, and $v$ is called a *sub-exponential* random variable if $\|v\|_{\psi_1} \leq K_1$ for a constant $K_1$. In the literature, $\|v\|_{\psi_2}$ is referred to as the *sub-Gaussian norm* and $\|v\|_{\psi_1}$ is referred to as the *sub-exponential norm*. Note that, ignoring constants, sub-exponential tails decay at $\exp(-t)$ whereas sub-Gaussian tails decay as $\exp(-t^2/2)$ so that sub-exponential tails are heavier than sub-Gaussian tails.

The following result will be used in our analysis:

**Lemma 9** *Let $v_i, v_j$ be sub-Gaussian random variables with $\max\{\|v_i\|_{\psi_2}, \|v_j\|_{\psi_2}\} \leq K_2$. Then $v_i v_j - \mathbb{E}[v_i v_j]$ is a sub-exponential random variable with $\|v_i v_j - \mathbb{E}[v_i v_j]\|_{\psi_1} \leq 4K_2^2$.*

*Proof:*  By definition,

$$\|\mathbb{E}[v_i v_j]\|_{\psi_1} = |\mathbb{E}[v_i v_j]| \leq \mathbb{E}|v_i v_j| \leq \|v_i v_j\|_{\psi_1} . \qquad (62)$$

Using triangle inequality, we have

$$\|v_i v_j - \mathbb{E}[v_i v_j]\|_{\psi_1} \leq \|v_i v_j\|_{\psi_1} + \|\mathbb{E}[v_i v_j]\|_{\psi_1} \leq 2\|v_i v_j\|_{\psi_1} . \qquad (63)$$

Since $v_i, v_j$ are sub-Gaussian random variables, for any $p \geq 1$,

$$\mathbb{E}|v_i|^p \leq (K_2\sqrt{p})^p \quad \text{and} \quad \mathbb{E}|v_j|^p \leq (K_2\sqrt{p})^p . \qquad (64)$$

Then, using Cauchy-Schwartz inequality

$$\mathbb{E}|v_i v_j|^p = \mathbb{E}|v_i|^p|v_j|^p \leq \left(\mathbb{E}|v_i|^{2p}\mathbb{E}|v_j|^{2p}\right)^{1/2} \leq \left((K_2\sqrt{2p})^{2p}(K_2\sqrt{2p})^{2p}\right)^{1/2} = K_2^{2p}2^p p^p .$$

Hence,

$$\|v_i v_j\|_{\psi_1} = \sup_{p \geq 1} p^{-1}(\mathbb{E}|v_i v_j|^p)^{1/p} \leq 2K_2^2 .$$

The result then follows from (63). ∎

We also need the following Bernstein-type inequality for sums of independent sub-exponential random variables [2]:

**Theorem 4** *Let* $v_1, \ldots, v_n$ *be independent centered sub-exponential random variables, and* $K_1 = \max_i \|v_i\|_{\psi_1}$. *Then, for every* $\mathbf{b} = (b_1, \ldots, b_n) \in R^n$ *and every* $t \geq 0$, *we have*

$$\mathbb{P}\left\{\left|\sum_{k=1}^{n} b_k v_k\right| \geq t\right\} \leq 2\exp\left\{-c_0 \min\left(\frac{t^2}{K_1^2\|\mathbf{b}\|_2^2}, \frac{t}{K_1\|\mathbf{b}\|_\infty}\right)\right\}, \tag{65}$$

*where* $c_0 > 0$ *is an absolute constant.*

We will be also using the following form of the above result:

**Corollary 1** *Let* $v_1, \ldots, v_n$ *be independent centered sub-exponential random variables, and* $K_1 = \max_i \|v_i\|_{\psi_1}$. *Then, for every* $\epsilon \geq 0$, *we have*

$$\mathbb{P}\left\{\left|\frac{1}{n}\sum_{k=1}^{n} v_k\right| \geq \epsilon\right\} \leq 2\exp\left\{-c_0 \min\left(\frac{\epsilon^2}{K_1^2}, \frac{\epsilon}{K_1}\right)n\right\}, \tag{66}$$

*where* $c_0 > 0$ *is an absolute constant.*

Next, we consider perturbing the covariance matrix $\hat{\Sigma}$ using independent zero-mean sub-exponential random variables. First, we illustrate that the nature of the tail bounds stay unchanged under such perturbations. Then, we show that one can do deterministic perturbations to get coarser and/or truncated representations of the sample covariance matrix, saving on the memory foot-print of the covariance matrix without affecting the statistical guarantees.

Let $\Delta_{ij}$ be independent zero mean sub-exponential random variables, and we consider the modified covariance matrix with entries:

$$\hat{S}_{ij} = \frac{1}{n}\sum_{k=1}^{n} R_{ik}R_{jk} - \frac{1}{n}\left(\frac{1}{n}\sum_{k=1}^{n} R_{ik}\right)\left(\frac{1}{n}\sum_{k=1}^{n} R_{jk}\right) + \Delta_{ij}. \tag{67}$$

Then, we have the following result:

**Theorem 5** *Let* $K_2 = \max_i \|R_{i\cdot}\|_{\psi_2}$ *and* $K_1 = \max_{ij} \|\Delta_{ij}\|_{\psi_1}$. *Assuming* $K_1 \leq 4K_2^2$, *we have*

$$\mathbb{P}\left\{\max_{ij} |\hat{S}_{ij} - \Sigma_{0,ij}| \geq \epsilon\right\} \leq 6\exp\left\{-c_0 \min\left(\frac{\epsilon^2}{36c_1^2 K_2^4}, \frac{\epsilon}{12c_1 K_2^2}\right)n\right\}, \tag{68}$$

*for suitable positive constant* $c_0, c_1$.

*Proof:* By definition, for any $i, j$,

$$\mathbb{P}\left\{|\hat{S}_{ij} - \Sigma_{0,ij}| \geq \epsilon\right\}$$

$$= \mathbb{P}\left\{\left|\left(\frac{1}{n}\sum_{k=1}^{n} R_{ik}R_{jk} - \Sigma_{0,ij}\right) + \Delta_{ij} - \frac{1}{n}\left(\frac{1}{n}\sum_{k=1}^{n} R_{ik}\right)\left(\frac{1}{n}\sum_{k=1}^{n} R_{jk}\right)\right| \geq \epsilon\right\}$$

$$\leq \mathbb{P}\left\{\left|\frac{1}{n}\sum_{k=1}^{n} R_{ik}R_{jk} - \Sigma_{0,ij}\right| \geq \epsilon/3\right\} + \mathbb{P}\left\{|\Delta_{ij}| \geq \epsilon/2\right\} \tag{69}$$

$$+ \mathbb{P}\left\{\left|\frac{1}{n}\left(\frac{1}{n}\sum_{k=1}^{n} R_{ik}\right)\left(\frac{1}{n}\sum_{k=1}^{n} R_{jk}\right)\right| \geq \epsilon/3\right\}$$

where the last inequality follows from the union bound. Each term in the summation considers a large deviation bound for a sub-exponential random variable. For the first term, from Lemma 9, $K_{1,1} = \|R_i R_j - \mathbb{E}[R_i R_j]\|_{\psi_1} \leq 4K_2^2$. For the second term, from the assumption regarding $\Delta_{ij}$, $K_{1,2} = \|\Delta_{ij}\|_{\psi_1} \leq 4K_2^2$. Now, we focus on the third term. Recall that the sub-Gaussian norm of the sum of sub-Gaussian random variables satisfy the following inequality [2]:

$$\left\|\sum_{k=1}^{n} R_{ik}\right\|_{\psi_2}^2 \leq c_1 \sum_{k=1}^{n} \|R_{ik}\|_{\psi_2}^2, \tag{70}$$

for an absolute constant $c_1$. In our context, since $\|R_{ik}\|_{\psi_2} \le K_2$, we have

$$\left\| \sum_{k=1}^{n} R_{ik} \right\|_{\psi_2} \le \sqrt{c_1 n} K_2 \quad \Rightarrow \quad \left\| \frac{1}{n} \sum_{k=1}^{n} R_{ik} \right\|_{\psi_2} \le \sqrt{\frac{c_1}{n}} K_2 \le \sqrt{c_1} K_2 \, . \tag{71}$$

From Lemma 9, we have

$$K_{1,3} = \left\| \left( \frac{1}{n} \sum_{k=1}^{n} R_{ik} \right) \left( \frac{1}{n} \sum_{k=1}^{n} R_{jk} \right) \right\|_{\psi_1} \le 4 c_1 K_2^2 \, . \tag{72}$$

Then, considering all three terms, using Corollary 1 for the first two terms and Theorem 4 for the third term, we have

$$\mathbb{P}\left\{ |\hat{S}_{ij} - \Sigma_{0,ij}| \ge \epsilon \right\}$$

$$\le 2 \exp\left\{ -c_0 \min\left( \frac{\epsilon^2}{9 K_{1,1}^2}, \frac{\epsilon}{3 K_{1,1}} \right) n \right\} + 2 \exp\left\{ -c_0 \min\left( \frac{\epsilon^2}{9 K_{1,2}^2}, \frac{\epsilon}{3 K_{1,2}} \right) n \right\}$$

$$\quad + 2 \exp\left\{ -c_0 \min\left( \frac{\epsilon^2 n^2}{9 K_{1,3}^2}, \frac{\epsilon n}{3 K_{1,3}} \right) \right\}$$

$$\le 4 \exp\left\{ -c_0 \min\left( \frac{\epsilon^2}{36 K_2^4}, \frac{\epsilon}{12 K_2^2} \right) n \right\} + 2 \exp\left\{ -c_0 \min\left( \frac{\epsilon^2 n}{36 c_1^2 K_2^4}, \frac{\epsilon}{12 c_1 K_2^2} \right) n \right\}$$

$$\le 6 \exp\left\{ -c_0 \min\left( \frac{\epsilon^2}{36 c_1^2 K_2^4}, \frac{\epsilon}{12 c_1 K_2^2} \right) n \right\} \, .$$

That completes the proof. ∎

In particular, for sufficient number of samples such that $c\sqrt{\log p / n} \le 3 c_1 K_2^4$, we have

$$\mathbb{P}\left\{ \max_{ij} |\hat{S}_{ij} - \Sigma_{0,ij}| \ge c\sqrt{\log p/n} \right\} \le 6 \exp\left\{ -\frac{c^2 c_0}{36 c_1^2 K_2^4} \log p \right\} \le 6 p^{-c_3} \, , \tag{73}$$

where $c_3$ is a suitable constant. Note that the above corresponds to the result discussed in the main text.

A special case of such perturbations arise by choosing constant $\Delta_{ij}$ for each $(i, j)$ with $|\Delta_{ij}| \le c\sqrt{\frac{\log p}{n}}$ in order to truncate or coarsen entries in the sample covariance matrix. In particular,

(i) if $|\hat{\Sigma}_{ij}| \le c\sqrt{\frac{\log p}{n}}$, then it can be safely truncated to 0; and

(ii) numeric representation of any $\hat{\Sigma}_{ij}$ can be coarsened to the level $c\sqrt{\frac{\log p}{n}}$, e.g., one can rewrite

$$\hat{\Sigma}_{ij} = 1.29 \underbrace{317542365}_{\le c\sqrt{\frac{\log p}{n}}} \qquad \text{as} \qquad \hat{S}_{ij} = 1.29$$

without affecting the statistical properties of the estimated precision matrix $\hat{\Omega}$. Such truncation and coarsening can lead to significant savings in the memory foot-print of the sample covariance matrix.