[Reviews · NeurIPS 2013]

Submitted by Assigned_Reviewer_2

The authors discuss the problem of sparse precision matrix estimation using CLIME and present a scalable variant CLIME-ADMM along with a distributed framework for computations. Empirical results comparing the CLIME-ADMM algorithm to state-of-the-art techniques such as DC-QUIC, Tiger and Flare are presented.

Notation remains scattered in various parts of Section 1, 2 and 3. It may help instead to have a table with all the notations for ease of readability. For example, \rho and \eta do not seem to be defined in the text.

The block cyclic data distribution appears to be interesting- it is however not obvious why this scheme would achieve load balance and scalability. There are also no empirical results to prove this point.

It appears that the algorithm currently sets the column block size on an adhoc basis -- is there some intuition or theory to guide the choice of k?
A more general concern is that this is perhaps not a 'distributed' algorithm as no explicit mechanism of message passing and communication are discussed. Perhaps it would make sense to consider this a parallel implementation.

Minor comments:
Section 1: ' where \lambda is a tunning parameter -> Replace with 'tuning'
Section 5.1: 'As Tiger is parameter tunning free' -> Replace with 'tuning'
Section 2: 'CLIME is summerized in' -> 'CLIME is summarized in'

Summary: A large scale parallel algorithm for estimation of sparse precision matrix using CLIME is presented. The novelty involves the estimation of the precision matrix by column blocks, instead of column-by-column. Empricial analysis using OpenMPI and Scalapak is presented.


Submitted by Assigned_Reviewer_6

This authors proposed an inexact alternating direction method of multiplier (ADMM) for solving the linear program problem in the CLIME estimator. A convergence rate of O(1/T) is established. The algorithm is implemented using both shared-memory and distributed-memory architectures. Numerical comparisons with other methods, such as DC-QUIC, Tiger and CLIME-Flare, are provided.


Quality: The paper is technically sound.

minor comments:
1) The derivation of the inexact ADMM for CLIME is standard. See, for example:
http://math.nju.edu.cn/~hebma/English%20Version.htm

2) Due to the structures of CLIME, the solution of each subproblem can be computed componentwise and the most expensive operation is matrix-matrix multiplication. Hence, ADMM can be parallelized ideally for CLIME. It is well known that matrix-matrix multiplications can be parallelized well.

3) The set up of the numerical experiments can be further specified. For example, what is the accuracy achieved by each solver?

Clarity: The paper is well-organized.

Originality: The evaluation of ADMM in shared-memory and distributed-memory architectures is interesting.

Significance: Solving large scale sparse precision estimation problems is challenging. A simple yet robust distributed algorithm is helpful.
Summary: The evaluation of ADMM in shared-memory and distributed-memory architectures is interesting. The derivation of inexact ADMM is standard and the good scalability is due to the simple structure of CLIME.

Submitted by Assigned_Reviewer_7

Summary of the paper

This paper is concerned with the resolution of covariance selection problems in a very large scale setup (up to millions of features and trillions of parameters). The estimator considered is the CLIME, which is known to provide an estimator of the precision matrix of a Gaussian vector that can be solved column by column, thus more easily amenable to distributed computations. Theoretical analysis is also claimed to be easier than for the Graphical-Lasso. The authors developed a new algorithm to fit the CLIME by solving the problem block wise rather than column wise, couple to an inexact direction method of multipliers. Theoretical results on the convergence rate of this algorithm are stated, in term of both objective function and distance to optimality. Special attention is paid for special matrix structures (sparse and low rank) along the computation to reduce the computational burden. The algorithm is implemented within a scalable parallel computation framework for both shared and distributed memory systems. In the numerical studies, comparison are made in term of runtime with its direct competitors on both synthetic and real data. A detailed study concerning this new algorithm is also provided to evaluate the speedup regarding the block sizes and the number of cores on both shared memory and distributed memory systems.

Comments

This paper is well written and very clear. Novelty and contributions brought by the method are clearly stated in the introduction and connexion to existing works clearly established, with sounded bibliographical references. The algorithm is nicely introduced with a good balance between technical points of numerical analysis/algebra and theoretical guarantees for convergence rates. The numerical analysis is appropriate and the implementation of the algorithm shows very impressive performances.

My only concern is the availability of such a method: if the source code (mostly based upon open source libraries) is not made available to the community, their is no point in such a work.
Summary: A sound algorithmic paper which deals with the important practical problem of covariance selection when the dimension is very large (millions of variables). Such a powerful tool should be rewarded by publication if it is made available to the community.
Author Feedback

Author rebuttal: We thank all the reviewers for detailed and insightful comments. We will fix the typos, update the draft suitably to incorporate the feedback and address the concerns.

Review 1: (1) We will improve the readability on notations as suggested by the reviewer.

(2) Since the 1D distribution is not efficient for matrix multiplication, our discussion focuses on 2D block distribution in a 2D process grid. The block cyclic distribution helps the algorithm achieve load balance and scalability in the following three ways.

First, as the algorithm progresses, X becomes sparse, and the sparsity structure changes over time (section 3.1). A simple block distribution like dividing the matrix X evenly into large consecutive blocks may assign dense part to some processes and sparse part to other processes, leading to possible load imbalance. In contrast, the block cyclic distribution uses small nonconsecutive blocks, and thus can largely achieve load balance.

Second, we have two matrix multiplications, W = A'*X and U = A*W, where A' is the transpose of A (p*n) and p >> n. Assume A is 10000*100, X = 10000*200 and the process grid is 100*2. After the multiplication W = A'*X, W is 100*200. If dividing A,X into big consecutive blocks, the block size is 100*50 for A and 100*100 for X. After the multiplication, the block size of W(100*200) is 50*100. Only 4 processes have blocks while the other 16 processes will be not used in the next multiplication. On the other hand, if using block cyclic distribution and block sizes for A and X are 10*10, the block size of W(100*200) is 10*10. Therefore, each process has 2 blocks.

Third, if blocks can fit in the cache, it will significantly reduce cache misses in the matrix multiplication. As a result, block size should depend on cache/memory size, matrix structure, matrix size and process grid.

We did do some experiments in trying different block sizes on a small dataset when we chose the block size used in the paper. Due to the space limitations and the fact that the block cyclic distribution has been studied in the literature [5,11,15,16], we did not report the results in the paper.

(3) The choice of k is related to the block size and number of blocks in X. Assume the process grid is p*q and block size of X is r*c, if k is less than q*c, some processors will be idle. If k is so large that too many blocks crowd in a single machine, the competition for resources and communication among processes will increase. We have not seen any theory related to choosing the optimal block size and matrix size.

(4) While message passing and communication are not needed in a shared-memory architecture, parallel matrix multiplication in a distributed-memory architecture requires very skillful design in message passing and communication, particularly when block cyclic distribution is used [5,11,15]. As parallel matrix multiplication is a mature tool, we did not discuss details of message passing due to the space limitations. We will discuss parallel I/O for block cyclic distribution and message passing in matrix multiplication in detail in a longer version along with the release of the code.

Review 2: (1) Inexact ADMM has been studied in prior work [2,12,27]; but we establish an O(1/T) convergence rate for the objective, which has not been established for inexact ADMM in the literature. Our proof technique is also different from [12] and other proof techniques in the mentioned website. While [12] uses variational inequalities and assumes variables to lie in a closed convex set, our proof technique simply uses convexity and does not need such assumptions; in particular, in Eq. (8), X is not assumed to lie in a closed convex set.

(1,2) Otherwise, while the derivation of an inexact ADMM algorithm for CLIME is standard, there are further non-trivial subtleties in designing a corresponding distributed algorithm by converting the constrained optimization problem in CLIME into basic arithmetic calculations through the use of inexact ADMM. State-of-the art distributed algorithms based on ADMM [2,23] first introduce local variables for each process and then enforce consensus constraints. As the number of local variables and consensus constraints is often linear with the number of processes, the increase of memory use, communication and consensus among local variables can slow down the algorithm and substantially increase the complexity in designing and implementing an efficient distributed algorithm. In [23], Parikh and Boyd design a distributed ADMM algorithm by introducing local variables for each block. In contrast, although we did not intentionally design a distributed algorithm by introducing local variables as [2, 23], our extremely simple algorithm is inherently a distributed algorithm, which saves considerable effort in designing and coding a distributed system. Moreover, our framework has more flexibility in choosing block size and block distribution, which can achieve load balance and the efficient use of memory hierarchies. If applying our idea to [23], [23] will have a much simpler distributed algorithm, which does not require additional local copies and consensus constraints but allows to choose block size and block distribution. The idea of [23] may not be suitable for equality constraint based on matrix-matrix multiplication since [23] considers an equality constraint based on matrix-vector multiplication and introduces local variables and consensus constraints.

(3) For ADMM-Clime, the stopping condition for optimality residuals is 0.005^2. Flare-ADMM uses 10^(-2) (stopping conditions may be different from ours) since it is slow.

Review 3: We thank the reviewer for encouraging comments. We will make the code public after the paper is published.